



# Ocean bubbles under high wind conditions. Part 1: Bubble distribution and development

Helen Czerski[1], Ian M. Brooks[2], Steve Gunn[3], Robin Pascal[4], Adrian Matei[1], Byron Blomquist[5,6]

[1]Department of Mechanical Engineering, University College London, London, WC1E 7BT, UK

[2]School of Earth and Environment, University of Leeds, Leeds, LS2 9JT, UK

[3]Department of Electronics and Computer Science, University of Southampton, SO17 1BJ, UK

[4]National Oceanography Centre, Southampton, SO14 3ZH, UK

[5]Cooperative Institute for Research in Environmental Sciences, University of Colorado, Boulder, CO, USA

[6]NOAA Physical Sciences Laboratory, Boulder, Colorado, USA

*Correspondence to*: Helen Czerski (h.czerski@ucl.ac.uk)

**Abstract.** The bubbles generated by breaking waves are of considerable scientific interest due to their influence on air-sea gas transfer, aerosol production, and upper ocean optics and acoustics. However, a detailed understanding of the processes creating

deeper bubble plumes (extending 2-10 metres below the ocean surface) and their significance for air-sea gas exchange is still lacking. Here, we present bubble measurements from the HiWinGS expedition in the North Atlantic in 2013, collected during several storms with wind speeds of 10-27 m s$^{-1}$. A suite of instruments was used to measure bubbles from a self-orienting free-floating spar buoy: a specialised bubble camera, acoustical resonators, and an upward-pointing sonar. The focus in this paper is on bubble void fractions and plume structure. The results are consistent with the presence of a heterogeneous shallow bubble

layer occupying the top 1-2 m of the ocean which is regularly replenished by breaking waves, and deeper plumes which are only formed from the shallow layer at the convergence zones of Langmuir circulation. These advection events are not directly connected to surface breaking. The void fraction distributions at 2 m depth show a sharp cut-off at a void fraction of $10^{-4.5}$ even in the highest winds, implying the existence of mechanisms limiting the void fractions close to the surface. Below wind speeds of 16 m s$^{-1}$ or $R_{Hw}= 2\times10^{6}$, the probability distribution of void fraction at 2 m depth is very similar in all conditions, but

increases significantly above either threshold. Void fractions are significantly different during periods of rising and falling winds, but there is no distinction with wave age. There is a complex near-surface flow structure due to Langmuir circulation, Stokes drift, and wind-induced current shear which influences the spatial distribution of bubbles within the top few metres. We do not see evidence for slow bubble dissolution as bubbles are carried downwards, implying that collapse is the more likely termination process. We conclude that the shallow and deeper bubble layers need to be studied simultaneously to link

them to the 3D flow patterns in the top few metres of the ocean. Many open questions remain about the extent to which deep bubble plumes contribute to air-sea gas transfer. A companion paper (Czerski, 2021) addresses the observed bubble size distributions and the processes responsible for them.



## 1 Introduction

The bubbles generated by breaking waves are an important feature of the ocean surface. They have a significant influence on air-sea gas transfer (Farmer et al., 1993; Wanninkhof and Triñanes, 2017; Goddijn-Murphy et al., 2016; Deike and Melville, 2018), sea salt aerosol production (Salter et al., 2014; Lewis and Schwartz, 2013; Gantt and Meskhidze, 2013; Norris et al., 2012), and the acoustical (Deane and Stokes, 2010; Deane, 2016; Ainslie, 2005) and optical properties (Terrill et al., 2001; Salisbury et al., 2013) of the upper ocean. Bubbles are thought to contribute to surfactant scavenging and the formation of the

sea-surface microlayer (Wurl et al., 2011). There have also been suggestions that bubble adsorption of surface-active carbohydrates may be important for the formation of transparent exopolymer particles (Zhou et al., 1998), and bubble processes are thought to influence turbulence and energy dissipation close to the ocean surface (Gemmrich and D.M.Farmer, 2004; Deike et al., 2016). However, these conclusions have mostly been drawn from the strong correlations between bubble presence and these processes in nature and in the laboratory, and the mechanistic details remain an active area of study. The links between

the immediate near-surface bubbles formed as a wave breaks and the deeper plumes observed over longer time periods are particularly unclear. It has been suggested that the deeper plumes are formed by advection of smaller bubbles downwards rather than being the direct consequence of a breaking waves (Zedel and Farmer, 1991; Thorpe et al., 2003), but there has been little in situ data available to explore those processes. This question is very relevant to the uptake of less soluble gases like oxygen, as well as for acoustical and optical studies. In addition, there is no clear picture connecting bubble formation

processes, advection, and the process and location of bubble termination. This is the first of two papers describing detailed bubble studies from the 2013 High Wind Speed Gas Exchange Study (HiWinGS) which aim to clarify some of these outstanding issues. This paper will consider bubble presence and movement tracked using bubble void fractions, and the second paper (Czerski, 2021) will address the mechanisms generating the observed structure of these plumes by considering bubble size distributions.

It is worth noting some fundamental points that sometimes cause confusion in discussions of this topic. The first is that the scientific need is usually to understand bubble *flux*: the rates at which bubbles are formed, changed, and destroyed. However, almost all open-ocean measurements to date are of bubble *presence*. Without information about lifetimes and the processes changing the bubble population, it is not possible to link bubble presence by itself to gas flux or particle production. A focus on the balance between bubble sources and sinks, rather than presence alone, is essential for progress. The second point is that

the highly heterogeneous distribution of near-surface bubbles is often described in terms of plumes (and indeed, we follow this convention here). However, a "plume" is a poorly constrained entity. The measured edges often depend on sonar settings or arbitrary thresholds, chosen for practical expediency rather than being based on clearly defined, physically meaningful criteria. Deane (Deane, 2016) distinguishes between "plumes" and "clouds" - the immediate high void fraction region just after a wave breaks and the remnants after several seconds - but this is not standard nomenclature. As the data presented here shows, there

is a heterogeneous bubble field with robust statistical properties, and so while we use the word "plume", we note its limitations.

### 1.1 Background



Wave breaking and bubble production generally begin when the wind speed rises above 5-10 m s$^{-1}$ (Banner and Peregrine, 1993; Gemmrich and D.M.Farmer, 1999), although they can appear at wind speeds as low as 3 m s$^{-1}$ (Asher and Wanninkhof, 1998). A universal description of wave breaking and bubble production in the open ocean is still lacking. The primary motivation for further research is the need to improve parametrizations of air-sea gas transfer, particularly for carbon dioxide (Wanninkhof, 2014), but also oxygen (Chiba and Baschek, 2010; Atamanchuk et al., 2020), and aerosol production (De Leeuw et al., 2011) for use in weather and climate models. The sensitivity of model outputs to the details of the subsurface bubble physics is still a matter of debate. However, many authors have identified a more complete knowledge of bubble presence and subsurface bubble physics as a necessary step in order to refine the current generation of models (Goddijn-Murphy et al., 2016; Blomquist et al., 2017; Gantt and Meskhidze, 2013).

There is a range of proven methods for detecting subsurface bubbles in the open ocean, although there is no universal technique that can cover the full range of observed bubble sizes, spatial patterns and timescales. Bubbles are highly compressible and so the most commonly-used methods are acoustical, either using sonar and acoustical backscatter (Thorpe, 1982; Trevorrow, 2003; Wang et al., 2011), or acoustical resonators (Farmer et al., 1998b; Czerski et al., 2011; Farmer et al., 1998a), or passive acoustics (Deane, 2012). In general, active acoustical methods are limited to smaller bubble sizes (below 1 mm radius) and lower void fractions (below 10$^{-4}$). For larger bubbles and higher void fractions, specialised bubble cameras have been developed (Stokes and Deane, 1999; Leifer et al., 2003; Al-Lashi et al., 2018). Other systems have also been trialled, including holographic detection (Talapatra et al., 2012) and optical scattering techniques (Randolph et al., 2014).

The current evidence suggests that highest void fractions (> 10$^{-3}$), which are associated with an actively breaking wave, are only found within a few tens of centimetres of the surface (Deane, 2016) or approximately match the scale of significant wave height $H_s$ (Anguelova and Huq, 2012; Callaghan et al., 2016), although there is limited direct evidence for this at sea in very high wind conditions. Once wave-breaking and the consequent turbulence have formed the initial bubble population, the bubble size distribution evolves rapidly under the influences of buoyancy and probably also dissolution (Deane and Stokes, 2002) but no new bubbles are formed. The largest bubbles will rise to the surface within a few seconds, forming visible whitecaps or bursting. Smaller bubbles remain in the water column, are advected by turbulent water flow, convection or Langmuir circulation (Trevorrow, 2003; Thorpe et al., 2003) and may either dissolve completely or eventually rise back to the surface. There is no consensus on how much of the initial population returns to the surface, and how the subsurface residence time probability distribution varies with bubble size. For practical reasons, studies either focus on processes very close to the surface (usually laboratory studies), or the deeper plumes observed at sea using upward-looking sonar, but rarely both at the same time.

## 1.2 Previous void fraction observations

Many studies have shown that small bubbles can form long-lasting plumes many metres in depth and with void fractions from 10$^{-8}$–10$^{-5}$. Goddijn-Murphy et al. (Goddijn-Murphy et al., 2016) noted that the bubble size contributing most to the void fraction



during active breaking is bigger than a millimetre, and Graham et al. (Graham et al., 2004) found that the bubble radius contributing most to void fraction in the deeper plumes is 100 μm.

Observations of these plumes have been made in a range of open ocean conditions (Vagle et al., 2010; Graham et al., 2004; Wang et al., 2011; Trevorrow, 2003), mostly using sonar data to record backscatter intensity. This technique gives good spatial information but cannot provide void fractions without additional information on bubble size distributions. Comparisons of these data sets can be difficult because the definition of the bubble plume "edge" depends on the sonar frequency and the thresholding techniques used, which are not always clearly stated. However, the measured plume depths have been found to vary with wind speed.

Bubble plumes are commonly described using two parameters: bubble penetration depth and an exponential decay constant that quantifies the decrease in bubble presence with distance from the surface (usually the only available measure is acoustical backscatter, so the decay constant actually refers to the decrease in scattering strength with depth). Reported plume depths vary from 5-25 metres. Wang $et$ $al$ (Wang et al., 2011) made measurements in the most extreme conditions in the literature, with wind speeds up to 50 m s$^{-1}$. They observed that the bubble plume depth increased linearly for wind speeds up to 35 m s$^{-1}$ (where the depth had reached 20 m) but increased more slowly after that, reaching approximately 25 m at 50 m s$^{-1}$. The exponential decay constant was found to be 0.8 m by Thorpe (Thorpe, 1982), 0.7-1.5 by Crawford & Farmer (Crawford and Farmer, 1987), and 0.5–3 m and strongly related to the plume depth by Trevorrow (2003). Graham et al. (Graham et al., 2004) observed that e-folding depth for acoustical backscatter was related to wind speed, varying from 0.2 m for very low winds (6 m s$^{-1}$) to 1 metre for higher winds (12 m s$^{-1}$) in 10 metre deep water. To our knowledge, the studies by Vagle et al. (Vagle et al., 2012; Vagle et al., 2010) are the only ones to use bubble size distribution measurements instead of backscatter as a basis for a parametrisation, producing bubble size distribution dependent estimates of bubble-mediated fluxes of $O_2$ and $N_2$. There is a huge amount of detail in many of these papers, but the difficulties in adequately describing all relevant parameters and in making backscatter comparisons mean that no universally-applicable description of these deep bubble plumes is agreed on. A full review of deep bubble plume measurements is beyond the scope of this paper, but an excellent review of the literature on this topic before 2004 can be found in Graham $et$ $al$ (Graham et al., 2004). Two important conclusions are that the penetration depth is most strongly related to wind speed, and that most bubbles come from shorter and steeper waves. A notable recent modelling study by Liang et al (Liang et al., 2012) suggests that the e-folding depth scales with friction velocity, and that wave age has a significant effect on plume behaviour.

**1.3 Bubble disappearance**





Considerable attention has been paid to bubble source functions, but there are far fewer studies on bubble sinks. Although bubble dissolution is routinely and accurately modelled for bubbles coated with surfactant monolayers in the lab environment (Azmin et al., 2012; Lozano and Longo, 2009), bubble dissolution in the ocean is significantly more challenging. This is because bubbles are stabilised by a complex mixture of surfactants (Frew et al., 1990; Wurl et al., 2011; Zhou et al., 1998) and are likely to be stabilised still further by particulates and gel-like materials on their surface. In one of the very few explicit

studies on this topic, Johnson and Wangersky (Johnson and Wangersky, 1987) showed that bubbles stabilized by a combination of surfactant and particulates would maintain their size for many hours, although a relatively small pressure increase could cause them to collapse very rapidly. It has generally been assumed that bubbles in the ocean dissolve gradually, but we are not aware of any direct evidence for this. Characterising the mechanism of bubble disappearance in the ocean is a necessary step towards understanding fluxes, and a considerable gap in current knowledge.


### 1.4 Influence of other parameters

The surface ocean is a complex environment with many features that are expected to have a second order influence on bubble production and their path through the water column. Some of these are the environmental conditions that affect breaking: the presence of swell, relative orientation of wind and waves, whether winds are rising or falling, and the surface heat flux.

Relevant water parameters such as temperature and surfactant presence must also be considered.

It is widely accepted that the existence of either soluble or insoluble natural surfactants in ocean water is likely to influence bubble production, bubble residence time, and the contribution of bubbles to air-sea gas transfer and aerosol production. Laboratory studies have made some progress on these questions (Callaghan et al., 2013; Callaghan et al., 2017; Anguelova

and Huq, 2012), but are hampered by the difficulty of conducting experiments with realistic natural surfactants. Characterising surfactants in the natural environment is a significant challenge.

Water temperature is expected to influence bubble processes at sea, but as yet there is no clear consensus on how. Studies by Salter et al. (Salter et al., 2014; Slauenwhite and Johnson, 1999) observed a significant increase in bubble production at lower

temperatures in laboratory experiments, and Salisbury et al (Salisbury et al., 2013) saw whitecap fraction decrease slightly as temperature increased in ocean satellite data. However, Callaghan (Callaghan et al., 2014) saw a 6% increase in air entrainment in wave tank experiments as water temperature was raised from 5°C to 30°C.

Other parameters reported to affect bubble formation, movement, and lifetime include phytoplankton presence (Kuhnhenn-

Dauben et al., 2008), surface heat flux, upper ocean stratification (Vagle et al., 2012), whether winds are rising or falling (Hwang et al., 2016), and wind-wave Reynolds number (Salisbury et al., 2013; Toba et al., 2006; Scanlon and Ward, 2016; Brumer et al., 2017a).

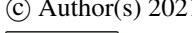



This wide range of potential influences and lack of field data complicates the current efforts to improve parameterisations of
ocean processes, a necessary step for improving weather and climate models, and to predict the biogeochemical consequences
of our changing climate (Talley et al., 2016). For example, carbon and oxygen uptake in the North Atlantic have been found
to be highly variable (Watson et al., 2009; Woosley et al., 2016) but the complexities are not yet understood. Improved
parameterisations of carbon dioxide and oxygen flux across the ocean surface are needed to forecast the future uptake of these
gases by the ocean. Bubble-mediated transfer is known to make a significant but poorly parametrized contribution in both
cases (Goddijn-Murphy et al., 2016; Atamanchuk et al., 2020) and a better mechanistic understanding has been identified as
critical for future improvements.

All the data presented here were collected during the HiWinGS expedition in 2013. This paper will give an overview of
HiWinGS and the measurement methods used, describe the observed bubble void fractions during stormy conditions, and
present evidence on the mechanisms of deep bubble plume formation. We will not address air-sea gas transfer directly, but
instead focus on bubble processes. The companion paper (Czerski, 2021) will present a separate analysis of the observed
bubble size distributions and the implications for bubble advection and destruction processes, along with a summary of
mechanistic understanding.

**2. Methods**

**2.1 HiWINGS expedition**

The overall aim of HiWinGS was to improve understanding of turbulent air-sea gas exchange processes in high wind conditions
by making direct measurements of the fluxes of trace gases and physical parameters, combined with sea state, wave, and
bubble physics (Brumer et al., 2017b; Kim et al., 2017; Blomquist et al., 2017; Yang et al., 2014).
The HiWinGS cruise took place between 9 October and 14 November 2013 on the R/V *Knorr* in the North Atlantic Ocean to
the south of Greenland, a region known as a significant sink for $CO_2$ and a period chosen to maximise the number and severity
of the storms encountered. An overview of the expedition and the instruments deployed can be found in Blomquist *et al*
(Blomquist et al., 2017) along with the major gas transfer results. Figure 1 shows the time series of the wind measurements
over the whole expedition, with periods when the buoy carrying bubble sensors was in the water highlighted.

The bubble measurements used here were made during four buoy deployments: 18-21 October (station 3), 24-27 October
(station 4), 1-4 November (station 6), and 7-10 November (station 7). The station numbers correspond to those in the Blomquist
(Blomquist et al., 2017) paper.  A summary of the conditions during each deployment is given in Table 1.

**2.2 Measurements**

The bubble instruments were all mounted on an 11-m long free-floating spar buoy. This was deployed in advance of each
storm, left to drift freely, and recovered after the storm. The buoy carried a bubble camera, acoustical resonators, an upward



looking sonar, wave wires, a downward-looking foam camera, inertial motion measurement unit, and an acoustic Doppler velocimeter (ADV). Figure 2 shows the position of the instruments on the buoy. A description of the buoy and its performance

can be found in (Pascal et al., 2011).

The two most significant characteristics of the buoy design relate to its buoyancy and its ability to orient into the wind. The buoy is split into two main sections: the main hull, a cylinder 25 cm in diameter and 6 m long which remained fully submerged at all times, and the top section, a cylinder 10 cm wide and 4 m long, which protruded through the sea surface. Above the top

section, a watertight dome carried the inertial measurement unit, additional small cameras, and a specialised foam camera. The base is formed of two hexagonal damping plates which carried four 24V Deepsea Light & Power batteries with a combined weight of 80kg, providing significant additional stability and reducing the natural vertical oscillation frequency of the buoy to a period of approximately 8 s. Small waves pass by without causing significant vertical motion, while the buoy rides over the larger swell. A lead ballast weight was suspended off-centre from the buoy base, forcing the buoy to sit in the water at an angle

of approximately 8 degrees to the vertical. Previous tests have shown that the hanging ballast has the effect of orienting the buoy into the wind (with the ballast on the downwind side), tracking the wind direction to within a few degrees (Pascal et al., 2011). The buoy was designed to keep the wave wires on the upwind side of the buoy, and all instruments except the ADV were also mounted on the upwind side. The ADV data shows that the wind forcing on the dome pushed the buoy downwind at a speed comparable to the near-surface Stokes drift, producing a complex flow profile relative to the buoy. A full description

is given in Appendix A. At the depths of the camera and resonator, the most likely situation is that both instruments were measuring in water which had flowed around the buoy from the downwind direction. However, given the constant vertical motion of the buoy and the turbulence in the surrounding water, we are confident that both instruments were making measurements that were representative of the bulk water around them.

**2.2.1 Bubble camera & analysis methods**

The bubble camera was custom-built for this expedition; a detailed overview of its technical specifications and capabilities are given in Al-Lashi *et al* (Al-Lashi et al., 2018). The sample volume is defined by a light sheet, a concept developed successfully by Stokes & Deane (Stokes and Deane, 1999). The camera sat in a T-shaped housing with a circular window that faced into the oncoming waves. Strobe light sheets were projected forwards on four sides of the camera lens and were then deflected

using 45° mirrors to form a light sheet 5 mm thick and positioned a few centimetres in front of the camera lens. The duration of the strobe pulses was less than 5 µs, flashing at 15 Hz continuously throughout each recording period. A CCD camera with a telecentric lens was synchronised to each flash, and images were recorded at a resolution of 2048 x 2048 pixels with a square field of view 4 cm across. The minimum detectable bubble size was considered to be one dark pixel surrounded by 4-8 lighter ones (on a 3x3 grid), and tests demonstrated that the camera could reliably detect bubbles between 20 µm and 10 mm in radius.

Tests showed that the chosen design did not significantly interfere with the water flow in a way that would bias the





measurements, and analysis of the final data showed that the void fractions and bubble sizes observed were well below the thresholds where this would be a concern.

The camera was mounted as close to the surface as possible, which was 2 m below the waterline since the upper thin section of the spar could not support a heavy instrument. The total acquisition time available for each deployment was 9 hours. In order to follow the bubble plumes produced over 2-3 days as a storm passed overhead, acquisition was split into periods of 45 minutes which started every three or four hours, depending on the expected storm duration. Approximately 850,000 individual images were collected during the deployments described here, and an efficient algorithm for image processing was developed for automated analysis (Al-Lashi et al., 2016).


### 2.2.2 Acoustical Resonator

Bubbles are very responsive to incident sound waves, with a natural frequency strongly dependent on bubble radius. Acoustical resonators are robust devices which use active acoustics to detect detailed size distributions of small bubbles (5-500 μm in radius) with void fractions from $10^{-8}$ to $10^{-4}$. These resonators measure the acoustical attenuation of water, and in the absence

of other scatterers, broadband acoustical attenuation can be translated directly into bubble size distributions (Breitz and Medwin, 1989; Farmer et al., 1998a; Farmer et al., 2005; Czerski, 2012; Czerski et al., 2011). The resonators used here consisted of a pair of flat circular transducers with a diameter of 22 cm that face each other with a gap of 19 cm between them. The sample volume is the entire space between the plates, providing a bulk measure of acoustical properties. One transducer transmits white noise with a frequency range from 3 kHz - 1 MHz for 0.25 seconds and is then switched off for 0.75 s to allow

for data storage, producing one measurement every second. The other transducer records the data. The sharp peaks of the resonant frequencies are very clear in the Fourier transform of the output (Czerski et al., 2011), and are significantly attenuated by the presence of bubbles. A straightforward inversion algorithm can be used to convert the attenuation data into bubble size distributions (Czerski, 2012).

A pair of acoustical resonators was attached to the buoy, with nominal depths of 4 m and 6 m. Due to hardware issues, all the

data presented here is from the device at 4 m. Measurement periods were set to coincide with the camera acquisition, and total measurement time was limited by the battery to approximately 35 hours. A 4096 point FT spectrum was used and the effective measurement range for the data presented here was 6–173 μm radius. The measurements show that bubbles present at 4 m depth were highly unlikely to approach the upper limit of this range (Czerski, 2021).

### 2.2.3 Sonar

To provide context for the camera and resonator measurements, an upward-pointing Imagenex model 837A Delta T sonar, operating at 260 kHz, was deployed at the base of the buoy, eight metres below the waterline. The sonar was positioned so that one edge of the measurement arc touched the buoy hull, and the arc stretched outward from the buoy hull to scan a vertical slice through the water on the upwind side of the buoy, as shown in Fig. 2. The device ran directly from a PC installed in a



waterproof housing at the base of the buoy. Measurements were made continuously from deployment until the battery ran out, which varied from 26-36 hours. The sonar recorded six complete scans per second, which were averaged before analysis to form 1s images.

Bubble plumes were clearly visible on the sonar images, and the position of the camera and resonators within bubble plumes could be monitored. The unique benefit of combining the sonar with camera and resonator measurements is the possibility of
acquiring detailed bubble size distributions within a sonar-measured plume, providing a direct link between the bubble size distributions and the spatial extent of the plume within which they sit. We note that the sonar frequency of 260 kHz would cause a resonant acoustical response in bubbles of 12.5 microns in radius, and is therefore particularly sensitive to bubbles of that size. There is some evidence that during periods when bubble plumes were visible on the sonar but not on either camera or resonator, the number of bubbles between 11 and 14 μm was higher than in other periods, suggesting that at least some of
the deep bubble plumes seen in sonar images both in this experiment and in previous experiments were due to small numbers of resonant bubbles (void fraction $O(10^{-9})$), rather than a significant total void fraction.

### 2.2.4 Auxiliary data streams

A Nortek Vector acoustic Doppler velocimeter (ADV) was attached to the buoy hull to provide 3D measurements of relative
flow velocity. Care is needed in interpreting these measurements because the ADV was positioned very close to the side of the hull (90° from the wind direction), but they allow a measure of the relative movement of the buoy through the water around it..

Three sets of wave wires (Pascal et al., 2011) ran parallel to the thin section of the spar buoy hull, passing through the waterline.
The total measurement length was four metres, and the buoy was ballasted with the wave wire halfway point at the waterline. Coupled with measurements of spar motion from the IMU, these provided detailed 1D wave data at the buoy itself, and allow the variability in the depth of bubble measurements below the wave surface to be made. Instantaneous depth data was found to fit a normal distribution during each deployment. The standard deviation in depth was a linear function of the 10m wind speed: $\sigma_{depth} = 0.16 + 0.028U_{10}$. At the highest wind speeds (above 20 m s$^{-1}$), the standard deviation is ~0.8, implying that the
bubble camera, at a mean depth of 2 m, was within 1 m of the surface approximately 10% of the time, and within 0.5 metres of the surface approximately 2.5% of the time. There were also small variations in mean hull depth, of order 10 cm, due to adjustments to the buoyancy between deployments.

A Waverider buoy (Datawell DWR-4G Waverider buoy, 0.4 m diameter) was deployed for the duration of each storm, providing 2D wave spectra. This buoy drifted freely, so measurements were not perfectly co-located with the spar buoy, but
the maximum separation after a large storm was only a few kilometres. This data was used to characterise the 2D wave field. 43 CTD casts were made during the cruise, with at least one each day except for the four days when the ship was in transit to the Gulf Stream (4–7 November). The CTD also carried a dissolved oxygen sensor. A WET Labs Chlorophyll WETSar fluorometer continuously sampled the ship's saltwater intake from five metres below the waterline. An overview of the





extensive set of wider environmental parameters measured from the ship is provided in Blomquist *et al* (Blomquist et al., 2017).

## 3. Results

### 3.1 Overview

The simplest measure of bubble presence at a single point is void fraction, the ratio of air to water volume. Figure 3 shows the
void fraction measured simultaneously by bubble camera and resonator, and compared with sonar backscatter for a 45-minute period on 2 November, during wind speeds of 18 m s$^{-1}$. The patterns in time match very well, and provide confidence that all systems were working as intended. Measured void fractions from the camera and resonator are shown on both linear and logarithmic scales to emphasise the point that a measurable bubble void fraction was present at 2 m depth at all times, even though the linear plots show distinct peaks. At 4 m, the void fraction is below the detection limit ($10^{-8}$) of the resonator almost
all the time. This suggests that the sonar used may also have had a detection limit of approximately $10^{-8}$, or possibly that bubbles were entirely absent at 4 m between the distinct plumes. We note that the bubble size ranges sampled by the resonator and camera are very different (radii of 6–173 µm and 20–4000 µm respectively). Extrapolating the observed bubble size distribution to 1 µm in radius suggests that the maximum possible contribution to the total volume from bubbles below 20 µm is extremely small (significantly less than 1%), while the small number of bubbles larger than ~100 µm reaching the depth of
the resonator make a negligible contribution to the total void fraction; a direct comparison between the two instruments is thus appropriate.

We observe that although the major features on all three timelines match, plumes are visible on the sonar at times when there are very low void fractions at 2 m and 4 m depth. This is a consistent feature and shows the limitations of sonar data in isolation. Significant backscatter can be caused by small resonant bubbles, but this does not necessarily indicate an equally significant
void fraction in those plumes. We note that the linear scale on the y-axis exacerbates this effect on this plot, but it highlights the care needed not to over-interpret bubble plumes observed by sonar alone. It is also difficult to make a direct quantitative comparison between the sonar backscatter at 2 m and the measured camera void fraction. The plume patterns agree well, but the regions of highest void fraction (the centres of big plumes) may be acoustically shielded (Deane, 2016) by the smaller bubbles around them, producing a low sonar backscatter signal for some high void fraction regions.
Figure 4 shows 1-minute averages of void fraction for station 6 (1–4 November). For most intervals there is no strong correlation between bubble presence at 2 m and at 4 m. Some of this may be due to plume shearing (see Sect. 3.3). We note that this seems to be a general feature and that although large identifiable plume structures exist that are connected vertically, they are overlaid on a significant general background level of bubbles which is highly heterogeneous and shows no strong vertical connection. Where the void fractions are correlated at both depths, the fitted line has a slope of 2, implying an e-
folding distance of 0.5 m (see Sect. 3.3.3 for further analysis of e-folding depths).

### 3.1.1 Buoy movement: Horizontal advection



The spar buoy was not stationary within its local water mass and the relative flow is highly depth-dependent. The water motion is due to a combination of Stokes drift, wind forcing on the exposed top of the buoy and wind-driven surface currents. As discussed in Appendix A, the downwind drift of the buoy dominated the surface-driven currents at all times at 4 m, and is likely to have dominated at 2 m. At the ADV depth the range of relative speeds between the buoy and surrounding water was 0.02–0.15 m s$^{-1}$ for wind speeds between 7 and 28 m s$^{-1}$, as the buoy drifted downwind. At a typical relative speed of 0.11 m s$^{-1}$, a single plume with a horizontal extent of 4 m perpendicular to the breaking crest would be crossed by the buoy in 36 s. Previous authors (Trevorrow, 2003) have reported that plumes may last for 20-90 s. The advection complicates data interpretation, since we cannot distinguish between changes in observed bubble parameters due to evolution over time from those caused by spatial variation within the plumes.

## 3.2 Breaking Waves and deeper plumes

The buoy dome carried a foam camera recording the upwind ocean surface. Very few active breaking waves were seen in its field of view during the measurement periods, partly because data was only captured during daylight hours, but also because the short duration of active breaking limits the probability of an event falling within the field of view. None of the observed breaking waves were directly correlated with bubble presence at 4 m depth, which is consistent with deep plume formation being independent of wave breaking. Figure 5 shows one example where an active whitecap was observed by the foam camera and appeared to precede the formation of a deeper bubble plume.

Figure 5 shows that the observed breaking event generated a shallow plume of bubbles within the top 2 m. After about 40 s, during which time a few waves have passed, this shallow plume appears to join or form a deeper plume. The buoy was drifting downwind throughout this period. Our interpretation is that the buoy and the shallow plume drifted into a convergence region between two Langmuir cells, where bubbles have already accumulated, and so intercepted an existing deep plume. Figure 6 shows the corresponding water velocity data from the ADV at 3.8m. The dashed box represents the time period covered by Fig. 5, with a few minutes either side shown for context. There is a pronounced sideways flow feature as the deeper plume appears, followed by a period of flow in the opposite direction, which would be consistent with Langmuir cells at an angle to the wind. The vertical flow is almost all downward throughout this period, indicating a convergence zone. It seems likely that the breaking wave was independent of the Langmuir cell. The foam patch at the surface was observed to move slowly downwind relative to the buoy. Small long-lasting bubbles from this breaking wave are likely to have remained in the convergence zone, contributing to the persistent plume there. The combination of the ADV and sonar data provide convincing evidence that the deep plumes are due to the convergence of Langmuir cells, and are not directly connected to the breaking process.





### 3.3 Void Fraction

Void fraction data lacks the detail of bubble size distributions, but provides a straightforward measure to follow plume
evolution over time, and to analyse overall patterns of bubble presence and absence. At 2 m depth, bubble presence fell below
measurable levels (a void fraction of $10^{-8}$) for only 4% of the total data acquisition time. As shown in Fig. 3, structure is visible
on the log scale at all times at the camera depth and consequently categorising individual plumes is a difficult task and labels
implying plume presence or absence should be treated with care.

All the bubble camera data discussed below is an amalgamation from all four deployments, just over 23 hours of data. The
resonator data covers a longer period (52 hours over three deployments) but the void fraction at 4 m only rose above the
resonator noise level of $10^{-8}$ in 9% of all resonator measurements.

### 3.3.1 Void fraction probability distributions with wind speed and $R_{Hw}$

The wind-wave Reynolds number $R_{Hw} = {u_* H_s}/{v_W}$ is a dimensionless number introduced by Zhao & Toba (2001) which
combines a measure of the sea state (significant wave height $H_s$) with friction velocity ($u_*$) and water kinematic viscosity ($v_W$).
It has been successfully used to parametrise $CO_2$ gas transfer (Brumer et al., 2017a) and is better than wind speed alone at
explaining the variability of sea-spray aerosol flux (Norris et al., 2012; Yang et al., 2019). Here we directly evaluate the
relationship between $R_{Hw}$ and deeper bubble plumes, and investigate whether knowledge of surface processes alone is sufficient
to predict subsurface bubble behaviour.

Figure 7(a) and 7(b) show the probability distributions for void fraction at 2 m split by wind-wave Reynolds number (using
the wind sea component of the significant wave height $H_s$); Figs. 7(c) and 7(d) show the same data split by wind speed. The
probability distributions separated by wind speed have slightly broader peaks than those partitioned by $R_{Hw}$, with a peak void
fraction that generally increases with wind speed. The lowest Reynolds number groups all show sharp peaks centred on a
consistent void fraction of $10^{-7.1}$, with similar distributions. Above $R_{Hw}= 2 \times 10^6$ the peak moves to significantly higher void
fractions and the distribution becomes much broader. The distributions separated by wind speed show a steadier increase in
the peak void fraction as wind speed increases, although the distributions below 16 m s$^{-1}$ are very similar. The peak position
for 0-8 m s$^{-1}$ is slightly higher than that for 8-12 m s$^{-1}$, but we note that this distribution is based on far fewer data points (44
minutes of data versus 498 minutes). The notable feature of Fig. 7 is that below wind speeds of 16 m s$^{-1}$ or $R_{Hw}= 2 \times 10^6$ , the
probability distribution of void fraction is very similar in all conditions. Above either threshold, bubble presence at 2 m depth
increases significantly.



Figures 7(b) and 7(d) show a sharp cut-off in the distributions above a void fraction of $10^{-4.5}$, even at the highest wind speeds. This suggests that there is a strong limitation on the void fraction at this depth. Over the four deployments, there are only 116 one second measurements of void fractions between $10^{-4.5}$ and $10^{-4}$ (all in clusters covering a few seconds each), and six one-second measurements between $10^{-4}$ and $10^{-3.5}$. As discussed above, the instantaneous camera depth varied with buoy movement, so these measurements could represent shallower depths than 2 m. Even if void fractions greater than $10^{-4.5}$ can

briefly exist at 2 m depth, it seems clear that they cannot be sustained.

The available void fraction probability distributions for the resonator data cover a narrower range of conditions. The instrument noise level (measured in void fraction) was $1\times10^{-8}$ for station 3, $3\times10^{-8}$ for station 6, and $1.7\times10^{-7}$ for station 7; the fraction of data that rose above the noise for these deployments was 14%, 10% and 3%. Figure 8 shows the distribution of measured void

fractions for one Reynolds number range (which included 85% of all above-noise resonator measurements: 13119 data points). The normalisation uses the total measurement time for those data, including the 90% of time during which the resonator data fell below the noise level (not shown). The void fraction at 4 m very rarely rises above $1\times10^{-7}$ in wind speeds up to 19 m s$^{-1}$. The peak void fraction at 2 m is $10^{-6.92}$ and at 4 m it is $10^{-7.06}$, well above the noise threshold. The increase in pressure with depth is expected to reduce a void fraction of $10^{-6.92}$ at 2 m to $10^{-7.00}$ at 4 m if there is no loss of gas. This only accounts for

half of the peak offset although caution is required when interpreting the resonator peak, partly because of its irregular shape and partly because the data shown here came mostly from one period of a few hours. However, there seems to be no substantial change in peak void fraction with depth if the times when the void fraction is below the noise level are excluded. The similarity of the peak positions implies that bubbles are carried downwards in water packets that do not significantly mix with surrounding water before the bubbles are destroyed, and that the bubbles do not change their size due to slow dissolution before

destruction (in both these cases, a significant tail at the lowest void fractions would be expected, and this is not seen). This suggests that there is a relatively short (perhaps of the order of ten minutes) lifetime for small coated bubbles. If they could last for long periods (an hour or more) they would be expected to mix into the water column stochastically and produce regions with lower void fractions as the bubbles spread out in space.

**3.3.2 Rising and Falling Winds**

There has been sustained interest in the idea that the physical processes controlling gas flux and aerosol production depend on whether wind speeds are rising or falling (Liang et al., 2017; Dahl, 2003). Liang et al. (2017) estimated that the difference between air-sea gas fluxes at the same wind speed during rising and falling winds could be up to a factor of two. Differences

in gas transfer and bubble production between developing and fully developed seas have also been investigated (Blomquist et al., 2017; Scanlon and Ward, 2016; Clarke and Van Gorder, 2018). The distinction is important when making decisions between different possible parametrizations for these processes.





Figure 9(a) shows void fraction distributions separated by the trend in wind speed. If the hourly averaged wind speed was
lower or higher than the mean of the previous two hours by 0.5 m s⁻¹, the data from that hour were labelled "falling", or "rising"
respectively. Approximately one quarter of the data fell into each of those two categories. Otherwise data points were
considered ambiguous and are excluded. The included one-second void fractions from 2 m depth in each category were sorted
into 2 m s⁻¹ wind speed bins, and the median value of the void fraction in each bin has been plotted. Median void fraction
generally increases with wind speed, but the trends are different for rising and falling winds. For wind speeds below 11 m s⁻¹
and above 21 m s⁻¹, more bubbles were present during periods when the wind speed was rising. Between 11 m s⁻¹ and 21 m s⁻
¹, more bubbles were observed during periods of falling wind speed. The difference in void fraction is significant: of the order
of a factor of $10^{0.5}$ in the range 11-21 m s⁻¹.  Separation into "rising" and 'falling" in the highest wind speed range should be
treated with caution, because the wind was fluctuating within the top range for 8-9 hours, but mostly stayed within the 22-25
m s⁻¹ range.  At the extremes of the scale, there is also a bias in the recent wind history: the lowest rising wind speeds must
follow a period where winds were flat or falling, and the highest falling wind speeds must follow a period which was flat or
rising. For example, if the gas saturation state of the water increases during higher wind speeds, the average void fraction in
the 15 m s⁻¹ bin is a mean of the bubble presence at that wind speed in both higher and lower saturation states (following both
falling and rising winds), but the void fraction in the highest wind speed bin can never include the effects of a saturation state
from an even higher wind speed, and so will be biased low. Figure 9(c) shows the equivalent data from 4 m depth, and this
shows a similar but less pronounced pattern.

Three factors are expected to dominate bubble presence data: bubble production rate, bubble lifetime, and advection
mechanisms that determine whether or not bubbles reached the sensors. These effects cannot be clearly separated for our data.
Whitecap coverage (the best proxy for bubble production in the absence of detailed bubble measurements at the surface) can
be parametrized using wind and sea state (Brumer et al., 2017b), and the literature contains conflicting data on whether it
changes with rising/falling winds (Callaghan et al., 2008; Goddijn-Murphy et al., 2011). Langmuir circulation patterns are
thought to respond to the wind on timescales of a few minutes (Kukulka et al., 2010; Smith, 1992), so it seems unlikely that
changes in advection could be responsible for the asymmetry between bubble presence during rising and falling winds over
several hours. The most likely mechanism is therefore that bubbles can persist for significant periods after formation, and that
they last longer after periods of higher wind. An increase in local gas saturation levels following higher winds would explain
this, if bubble destruction mechanisms are strongly dependent on gas saturation state.

Figure 9(b) shows the void fractions split by inverse wave age (calculated as $u_*/c_p$, where $c_p$ is calculated using the whole sea
state rather than wind sea alone). It is notable that the same pattern is not apparent here, and that the void fractions for low and
high inverse wave ages are very similar. The void fraction variation appears to be dominated by whether wind speeds have
recently been higher or lower, rather than whether or not the wave field is fully developed with respect to the current wind
speed. Inverse wave age alone is therefore a limited proxy for bubble presence in the water column.



### 3.3.3 E-folding depths

As noted above, the observed plumes are highly inhomogeneous and so averaging over time can hide considerable complexity.
A common metric found in the literature is e-folding depth, the vertical distance over which the backscatter strength (or void fraction, depending on the dataset) decreases by a factor of $e$. We have the opportunity here to make this measurement for individual plumes, rather than from temporal and spatial averages. 20 plumes with void fraction pattern features that matched in time on both camera and resonator were analysed. These were the only unambiguous matching features in the 21 hours of simultaneous camera and resonator data, and many of these plumes were visible in the data for several minutes. Calculation
of the e-folding depth based on two points only (2-m and 4-m measurements) produced values of 0.3–0.6 m for all but one case; the only exception had a value of 0.94 m. There was no correlation between e-folding depth and the void fraction at 2 m. These values are on the lower end of measurements in the literature, and cover wind speeds of 12–17 m s$^{-1}$. No relationship was observed between e-folding depth and wind speed, although the relatively narrow wind speed range limits this comparison.

The e-folding depths were also calculated using 10 minute averages of the sonar backscatter at 2 m and 4 m depth, over the same wind speed range and the whole dataset. The vast majority were also tightly clustered between 0.3 and 0.6, suggesting that the averaged values of e-folding depth also hold for individual plumes.

### 3.4 Horizontal Structure

There are a total of 8 hours when both the camera and resonator data were of high quality and measuring simultaneously, during wind speeds between 14 m s$^{-1}$ and 21 m s$^{-1}$. In that time, there were 30 identifiable plume-like features measured at 4 m (several were close to the noise level, so the exact number is very sensitive to the threshold chosen), lasting between 30
seconds and 6 minutes. 20 of these matched up directly with plumes at 2 m, i.e. they were closely related in time and had the same shape. It is striking that the peaks at 4 m consistently lag those at 2 m by between 0 and 66 seconds (equivalent to a 0–9 m horizontal offset, given the buoy drift speed). Figure 10 shows an example of a typical event and the measured offset. The void fraction pattern is not symmetric in time — there is a sharp incoming rise and a slow tail-off at both depths.

All of the measured offsets bar one were positive (so bubbles were observed at 2m before 4m as the buoy drifted downwind), which suggests either that the plume shapes were consistently tilted towards the wind during the period of these measurements, or that plumes always become narrower with depth.





In general, the plumes seen in the sonar data are not symmetrical, and are often highly irregular in shape. Apart from the
offsets, there is no consistent discernible skew in plume shapes. Symmetrical triangular plumes are suggested by Zedel and
Farmer (Zedel and Farmer, 1991), but they process their data using the assumption that the plume is symmetric.

Figure 10(c) compares the horizontal offset with the ratio between the void fractions at 2 m and 4 m. It is notable that there are
no cases with both a very high void fraction ratio and a high horizontal offset. There are two possible interpretations of the
void fraction ratio between the two depths. The first is that when bubbles are advected downwards to form a plume, the initial
ratio is 1:1, and then bubbles are destroyed more quickly at depth. In this case, older plumes would have a higher ratio and the
ratio is driven by mechanisms acting at 4 m. The second is that only a small proportion of bubbles are ever advected
downwards, and that therefore high ratios are driven by intense bubble plumes near the surface which dissipate over time. In
this case, the ratio will decrease as the plume ages. Our observations are that the void fractions at 4 m only ever cover a narrow
range (as shown in Fig. 8), which points towards the second case. With this interpretation, Fig. 10(c) shows that the plumes
with the largest horizontal offsets seem to be older.

Figure 10(d) shows the wind speed at the time each horizontal offset was measured. The largest offsets were seen during
periods of the highest winds. This leads to one possible explanation: that the size of the Langmuir cells increases with wind
speed, and the offset is due to advection patterns that scale with the cell.

The offsets could also result from the shear associated with Stokes drift or other near-surface currents. The orbital motion
associated with a passing wave is almost circular, but also includes a small forward drift which is highest at the surface and
decreases with depth. However, to explain the direction of the consistent offset, this drift would have to be dominated by the
swell moving in the upwind direction rather than wind sea and wind-driven currents in the downwind direction. It is not
possible to be sure about the surface flows in this case, however it is thought that the Stokes drift from wind sea will dominate
that from at opposing swell at the surface but not at depth (Webb and Fox-Kemper, 2015). Figure 10(d) is hard to reconcile
with Stokes drift in the upwind direction due to opposing swell. There is no discernible pattern between the swell significant
wave height and the horizontal offset (not shown), although there were two major swells present at that time, as discussed in
Appendix A. We lack the data to provide a clear explanation for the consistent plume offset, given the uncertainty about the
current profile in the top few metres in these conditions. The offset matches what might be expected if the Stokes drift from
swell at this depth dominated the wind sea Stokes drift during this period, but the best available flow profile estimates from
the literature suggest that this is unlikely. However, the offsets were consistent and require further investigation.



A better understanding of the horizontal flow profiles could lead to a method for estimating the time since a plume formed. An assumption about the shape of the plume at its moment of "formation" would be needed, the simplest being that the plume is vertically aligned. If an estimate of the shear caused by near-surface currents was available, it would be possible to calculate

the time needed for the observed horizontal separation to be generated, and therefore the "age" of the deep plume. As discussed further below, this is not likely to be the time since the formation of the individual bubbles, but the time since the collection of bubbles was advected downward from the shallow bubble layer to form a deep plume. There is considerable uncertainty associated with applying this method here, but estimates using our data imply plume ages of the order of ten minutes. We note that bubble plumes were observed to shear by Crawford and Farmer (Crawford and Farmer, 1987), who also suggested that

the shape of a sheared plume might give an indication of age but did not attempt age estimates.

Sideways shear over long time periods may also explain the existence of plumes at 4 m that are not obviously connected to 2 m plumes. As the wind direction and speed change, the spatial distribution of plumes will constantly change as the local horizontal flow profile moves bubble plumes around. Changes to Langmuir circulation patterns have been observed to occur

over tens of minutes (Smith, 1992; Farmer and Li, 1994).

## 4. Discussion

### 4.1 Plume evolution

The model of plume development which is consistent with our data is as follows:

1. Breaking waves cause the formation of shallow bubble plumes, confined to the upper metre or so of the water column. The continual injection of bubbles into this layer produces a near-permanent bubble population with a probability distribution that depends on wind conditions and a structure that depends on both Langmuir circulation and near-surface shear currents. The maximum void fraction of approximately $10^{-4.5}$ observed at 2 m depth can be explained if

there are mechanisms acting in the shallow layer that limit the bubble population which can survive beyond the first minute or so after a breaking wave. However, this continuous layer appears not to extend to 4 m depth.

2. This shallow layer is advected sideways across the top of Langmuir circulation cells, or pushed across the cells by Stokes drift or wind-driven surface currents.

3. At the convergent limb of a Langmuir cell, water is advected downwards and if this contains bubbles from the shallow

layer, a deep plume is formed. These bubbles may already have existed for many tens of seconds before the "deep plume" formation event. At the downward speed shown in Fig. 6, bubbles could move from 2 m to 4 m depth in approximately one minute. However, this only happens at the locations that coincide with downward advection, which would explain why we did not observe bubbles at 4 m depth for 90% of the time and why, when they were present, the void fractions matched the 2 m measurements. The plume shape has several drivers, but it is likely to be

sheared by Stokes drift and wind-induced current shear.


4. Bubbles below depths of 4 m have a modest lifetime, which is consistent with the lack of a persistent background bubble presence at this depth. This model suggests that the deep plumes are continually fed from the shallow bubble layer and that the bubbles may be destroyed relatively quickly once they are advected downwards.

This picture explains the lack of any clear correlation between breaking waves observed at the surface and bubbles at 4 m. The shallow plumes are commonly observed in the sonar data under high winds and are superimposed on a persistent low and heterogeneous background population at 2 m. These results are consistent with observations made by D. Farmer and his collaborators over many years (Zedel and Farmer, 1991; Thorpe et al., 2003; Farmer and Li, 1994) which imply that there are two stages to deep plume formation.


The most likely explanation for the advection generating deep plumes is Langmuir circulation. This is consistent with Farmer & Li (Farmer and Li, 1994) who estimated that surface bubbles could be of order 100 s old before they reached 2 m depth in the downward flow of Langmuir circulations (at wind speeds of 10-15 m s$^{-1}$). Thus, bubble populations have significant time to evolve before the deep plumes are formed.


**4.2 Dependence on forcing conditions**

The bubbles present at 2 m depth show a clear dependence on the surface forcing, with void fraction increasing with wind-wave Reynolds number above a threshold of $R_{HW} = 2\times10^6$ or a wind speed of 16 m s$^{-1}$. It is interesting that the Reynolds

number threshold corresponds to that at which wind speed parameterizations of the $CO_2$ transfer velocity diverge (Brumer et al., 2017a), implying a change to bubble-mediated gas exchange. The void fractions in 2 m plumes are also higher during falling winds than rising, with some evidence for a similar pattern at 4 m.

The most likely explanation for the wind speed hysteresis appears to be changes to bubble destruction mechanisms during

rising and falling winds. It is likely that periods of increased wave breaking at higher wind speeds create surface waters with a higher concentration of oxygen and nitrogen. If the wind falls more quickly than the saturation state can adjust, bubbles produced during falling winds may last longer than those during rising winds because they are in water with higher gas saturation. A period of falling winds at high wind speeds implies that the recent winds were even higher, and the data support the idea that plumes of small bubbles last longer during these periods of (presumably) higher gas saturation.


Dahl (Dahl, 2003) saw increased acoustical scattering (assumed to be due to bubbles) when winds were falling compared to periods when they were rising, although these data were taken at low wind speeds (2–10 m s$^{-1}$). Liang (Liang et al., 2017) suggested that the transfer of oxygen and nitrogen into the ocean was greater when winds were rising rather than falling, and at 19 m s$^{-1}$ the flux of those two gases was almost doubled in the rising case compared with the falling case. They observe that





breaking waves are fewer in number but larger in size in developing seas, suggesting that this may cause bubbles to be entrained
to a greater depth, favouring gas transfer. This may leave a longer-lasting population once winds start to fall. It is also possible
that bubbles dissolve faster in rising seas, leaving fewer to be detected. This underlines the point that bubble presence may not
necessarily be directly related to gas flux in a simple way, since the presence of bubbles merely indicates that they have formed,
been advected to the measurement point, and have not yet been destroyed.

It is hard to separate out the effect of potential changes in bubble production and changes due to subsurface processes. Whitecap
observations may provide a first order proxy for bubble production. Whitecap fractions during HiWinGS were observed to
decrease as wave age increased (Brumer et al., 2017b), but Fig. 9 here shows no difference in observed void fraction at 2 m at
high and low wave ages. Callaghan (Callaghan et al., 2008) observed significantly higher whitecap coverage during periods
of falling winds compared with rising winds, for wind speeds between 10 and 24 m s$^{-1}$. However, the same separation was not

seen in satellite data during a later study (Goddijn-Murphy et al., 2011). These studies and our data do not provide a conclusive
steer on the likelihood of bubble production being directly affected by rising or falling winds, although the bubble presence at
2-m depth clearly is. However, the reversal of the pattern of void fraction with wind speed at the highest and lowest winds
(when there is a bias due to the most recent likely conditions) suggests that bubble lifetime is likely to be a more significant
influence than bubble formation on the shift in bubble presence during rising and falling winds.


In order to build a more complete picture of plume evolution, the two bubble layers (the shallow plumes in the top metre and
the deeper plumes which have been advected downwards) need to be studied simultaneously. Although the processes in both
cases are ultimately driven by similar phenomena – high winds causing breaking waves and contributing momentum to the

subsurface flow – the critical mechanisms for the two layers are likely to differ. This may explain why no simple links have
been found between deep bubble plumes and surface forcing conditions: very few measurements have been made in the top
metre to track the upper bubble field, and the deeper measurements have rarely had the auxiliary data needed to explain the
processes forming them. It is clear that data from different depths within both the shallow plumes and the deeper plumes,
correlated with flow and gas saturation data, is needed to follow the mechanisms driving our observations.


We have no direct measurements of the surfactant activity during HiWinGS. Chlorophyll was measured; levels were highest
during the first buoy deployment (approximately 1.5 $\mu$g l$^{-1}$) and generally decreased as the winter approached, reaching a low
of 0.3 $\mu$g/l in the last deployment (Table 1). However, recent studies (Sabbaghzadeh et al., 2017) suggest that there is no
universal relationship between chlorophyll levels and surfactant activity.


**5 Conclusions**





Direct measurements of bubbles in high wind conditions are relatively rare. HiWinGS offered an opportunity to combine several types of bubble measurement at wind speeds from 10-27 m s$^{-1}$. The results are consistent with a two-stage formation

mechanism for deep bubble plumes. Breaking waves form shallow bubble plumes which exist for tens of seconds but which remain in the top 1–2 m of the ocean. The continued action of breaking waves generated a continual bubble presence (with a void fraction greater than 10$^{-8}$) at 2 m depth with a probability distribution function that varied with wind speed. The probability distributions were observed to be very similar below wind speeds of 16 m s$^{-1}$ or $R_{\mathrm{Hw}} = 2 \times 10^{6}$, but changed significantly above those thresholds. The void fraction distribution at 2 m has a sharp cut-off at 10$^{-4.5}$ in all conditions, implying that this is the

maximum sustainable void fraction in the near-surface shallow bubble layer once the initial population has evolved through fragmentation and buoyancy.

If a shallow plume reaches a region where coherent advection patterns – assumed to be Langmuir circulation – have a downward limb, they are pulled down several metres on timescales of the order of a minute to form "deep" plumes. There are

thus two bubble layers: the near surface layer directly fed by breaking waves, and deeper plumes fed by coherent circulations. The processes generating the two layers are essentially independent, although both ultimately driven by wind stress. This two-stage process explains why breaking events on the surface (and the subsequent whitecaps) were not observed to correlate directly with deep bubble plumes.

The spatial distribution of bubbles within the top few metres is complex, and is dependent on several advection processes in addition to turbulence: Langmuir circulation, Stokes drift, and wind-driven near-surface shear currents. The 3D flow geometry could vary significantly with wind and wave conditions, and requires further study. Void fraction at 4 m only rose above the noise level for approximately 10% of the total measurement time. This implies a very limited lifetime for bubbles at this depth, perhaps of the order of a few minutes. For most wind speeds, more bubbles were present during falling than rising winds, and

no distinction is seen when the distributions are split by wave age. This suggests that wave age is limited as a proxy for bubble presence at depth, and trends in wind speed may be needed for parametrization.

Our data suggests that it is not straightforward to split the near-surface water into regions which either contain or do not contain bubbles. The void fraction probability distributions are smooth, although there are distinctive regions of high void fraction

which do correlate with deeper bubbles and which we identify as locations of deep plumes. We suggest that the void fraction distributions presented here are more useful than the number and frequency of plumes that cross a threshold intensity.

We found a horizontal offset and an asymmetry in bubble presence at 2 m and 4 m, with the plume edge consistently tilted towards the wind. We cannot offer a clear explanation for this observation, since there is considerable uncertainty in the likely

horizontal current profile during those events. It is possible that this offset could be used to estimate the elapsed time since deep plume formation, although robust calculations would only be possible with high resolution measurements of the near-



surface flow profile and a better understanding of the mechanisms operating as the plume structures are created and destroyed. We note that what were likely to be the oldest plumes were seen at the higher wind speeds, suggesting that bubbles may last longer in these conditions.


A critical question for future studies is the evolution and fate of the small bubbles in the shallow plumes which are not advected downwards. If they last for long periods, a high proportion may eventually dissolve completely into the ocean. In this case, the limiting step for oxygen uptake would be the formation process for these small bubbles. But if there are mechanisms within the shallow layer to bring them back to the surface so that they return gas to the atmosphere, the limiting step is the downward

advection to form deep plumes, which ensures that their contents must be taken up by the water.

For a more complete understanding of the processes involved, the results presented here should be combined with the detailed observations of bubble size distributions. These are addressed in a companion paper (Czerski, 2021).

**Appendix A**

The buoy was designed to keep the wave wires on the upwind side of the hull, to minimise the influence of the buoy itself on wave measurements. All bubble measurement instruments were also placed on the upwind side in the expectation that the dominant relative water flow would follow the wind. However, the water velocity data from the ADV show that this was not the case. A full analysis is beyond the scope of this paper, partly because the experimental data is limited and partly because

there is a lack of detailed knowledge concerning the response of the top few metres of the ocean to complex sea states. We discuss here our current understanding of our ADV data, the justification for the assumptions made in this paper, and recommendations for similar deployments in the future.

Figure A1(a) shows the ADV measurements of the relative water flow speeds past the buoy for different wind speeds. As

shown in Fig. A1(b), water flow relative to the buoy was always in the upwind direction with an offset of 0-20°, which suggests that the buoy was being pushed downwind by the wind. The response of the buoy to a given wind speed can vary considerably, but a linear fit to the data gives this relationship:

$$v_{ADV} = 0.0044u_{10} + 0.037 \quad , \tag{1}$$

where $v_{ADV}$ is the horizontal speed at which water is flowing past the buoy at the ADV depth in m s$^{-1}$.


An order-of-magnitude calculation shows that the observed speeds are consistent with the likely wind forcing on the buoy dome. A simple momentum equation estimate of the force on the cross-sectional area of the dome can be equated to the drag force on the hull from water flowing around it, as shown in equation 2.



$$\rho_{air} A_{dome} u^2 \;=\; \tfrac{1}{2}\, C_D\, \rho_{water} A_{hull} v_{drift}^2 \;, \qquad\qquad (2)$$

where $u$ is the wind speed at the dome height, $\rho_{air}$ and $\rho_{water}$ are the densities of air and water respectively, $C_D$ is the drag
coefficient for the hull (taken to be 1), $A_{dome}$ and $A_{hull}$ are the cross-sectional areas of the dome and the hull, and $v_{drift}$ is
the drift speed of the buoy relative to the water at its base. Using representative values for this buoy and taking the wind at 2
m using a logarithmic wind profile, this suggests $v_{drift} \sim 0.0085 u_{10}$ .

We conclude that the buoy was moving downwind due to the wind forcing on the dome. However, observations from the foam
camera clearly showed foam patches at the surface moving towards the buoy, and none were observed moving upwind relative
to the buoy. The resolution of this apparent conflict lies in the details of the wind-driven surface flows. These have been
described in various ways in the literature (Wu, 1983; Breivik et al., 2014; Laxague et al., 2018), but a recent paper (Van Der
Mheen et al., 2020) sets out three components: i) A surface layer several millimetres thick, which is dominated by viscous
effects, ii) a middle region a few metres thick, where the horizontal speed varies logarithmically with depth and iii) the Ekman
layer.  During the HiWINGS experiment, a typical Ekman depth was 150 m, and the calculated Ekman speeds varied very
little in the top 10 metres. Consequently we neglect it here although it will matter for the overall buoy drift speed, and we also
neglect the very thin viscous layer. There are two mechanisms driving the middle layer: the Stokes drift (caused by wave
motion) and wind-induced current shear.


Studies of the combined Stokes drift and wind-induced current shear have not yet provided a consensus on the surface flows
that are expected in different conditions, particularly when there are swells present which may have very different orientations
to the wind (Breivik et al., 2014; Morey et al., 2018; Clarke and Van Gorder, 2018).  There is a consensus that wind causes a
downwind surface flow (wind-induced shear current), with a directional offset of +/- 10° (Clarke and Van Gorder, 2018) and
a speed that decreases exponentially with depth with an e-folding depth of a few metres. The literature does not always
distinguish the Stokes drift and the wind-induced shear current, and we treat them as a combined phenomenon here. Webb &
Fox-Kemper (Webb and Fox-Kemper, 2015) demonstrate that using the assumption of a unidirectional sea (and therefore
ignoring wave-spreading and multidirectional waves) results in a significant overestimation (up to 70%) of the Stokes drift.
Their analysis also shows that higher frequency wind waves are likely to dominate close to the surface, and that lower
frequency swell is likely to dominate further down, and they make the point that a full spectral model is needed to analyse
each individual situation.

A wide range of wind and swell combinations were seen during the HiWINGS expedition. Figure A2 shows the angular offset
between wind and the dominant swell, overlaid with the periods when plumes were observed on both camera and resonator.
For the HiWINGS data, the issue of multidirectional waves is particularly relevant, because there was a significant swell at
180° to the wind during the November 1st-3rd deployment. Webb & Fox-Kemper (Webb and Fox-Kemper, 2015) analysed a




similar case in their data and found that their more complete (although not comprehensive) model showed that the opposing swell reduced the surface current by around 90%, with a counter-intuitive flow profile in which the surface speed was around half the value at 9 m depth. Figure A2 shows a notable shift of 180° in the relative swell direction at approximately 13:30 on

November 2nd. Figure A3 shows the full 2D wave measurements at that time. During this period, there is a jump in the dominant swell identified, and it can be seen that a significant opposing swell was present throughout that period. We expect that this will have substantially reduced the downwind surface current flow (Breivik et al., 2014; Webb and Fox-Kemper, 2015).


A full analysis of the likely flows relative to the buoy is beyond the scope of this paper, but we present a model which shows the features relevant during HiWINGS. Clarke & Van Gorder (Clarke and Van Gorder, 2018) propose a simplified model (equation 23 in that paper) for the Stokes drift in a realistic sea state which includes a factor to compensate for the overestimation seen by (Webb and Fox-Kemper, 2015). We apply it here using their estimate for e-folding depth for a

representative case: 15 m s$^{-1}$ winds, while ignoring the possible effects of swell. Figure A4 shows the combined effect of the wind-driven buoy drift (estimated as given above: $v_{drift} \sim 0.0085 u_{10}$ ) and the Stokes drift profile calculated from (Clarke and Van Gorder, 2018). This produces a profile which has the major features we observed: a constant upwind flow relative to the buoy at depth combined with a surface flow that can overtake the buoy.

The critical parameter for our experiment is the depth at which the net horizontal water flow relative to the buoy in the upwind-downwind direction is zero. Varying the parameters in this simple model to cover the range of our experiment suggests that the depth at which the relative flow shifts from downwind to upwind gets deeper as the wind speed increases and reaches nearly 2 m at 25 m s$^{-1}$. We therefore carried out our analysis on the assumption that both the bubble camera and the resonator were measuring bubbles which had travelled around the buoy from the downwind side at all times during HiWINGS. As

shown in Fig. 3 and as discussed in the main text, during the 1$^{st}$ – 3$^{rd}$ November deployment, the bubble plumes at 2 m and 4 m were highly correlated, which is consistent with this assumption. The bubbles therefore had to travel around the buoy to reach the sample volume, but given the dynamic flow situation caused by the surface currents, turbulence and the buoy motion, and the long lifetime of the plumes, it seems likely that sampling bubbles in water disturbed by the buoy will not have affected the measurements significantly.


We also note that the calculated flow profile cannot be responsible for the consistent offsets in plume position shown in Fig 10, because it shears in the opposite direction. However, we cannot rule out the possibility that at the highest winds the bubble camera was detecting bubbles coming from the upwind direction while the resonator was detecting bubbles from the downwind direction. Since there is no resonator data in the highest winds, we cannot make this comparison.


The only times during this expedition when we collected high quality bubble data from both 2 m and 4 m were all during periods with opposing swell, as shown in Fig. A2. Figure A1(c) shows the buoy drift speed only for periods when the wind was aligned with or opposite to the swell. At the lowest wind speeds, the drift speed is significantly lower when the swell was opposed to the wind although no clear separation is seen here at high wind speeds. We note that this plot takes no account of

the magnitude of the swell, only its direction, and that the periods with the highest winds did not have large opposing swells so we have no data for that condition.

Although the present analysis suggests that a situation similar to that shown in Fig. A4 is the most likely scenario for the

HiWINGS data, significant uncertainty about the details remains. This is particularly the case given the consistent plume offsets shown in Fig. 10, which could not be caused by the situation in Fig. A4, and which are currently unexplained. We have also neglected the possibility that the orientation of Stokes drift or of the plumes created by Langmuir circulation could produce a much more complicated geometry.

Two major recommendations arise for future measurement campaigns of this type. The first is to minimise the cross-sectional area of structures above the water surface and reduce the downwind drift of the buoy, unless that is a chosen design feature. The second is to collect data on the water flow profile relative to the buoy in detail, especially at the instrument depths and at the deepest point of the platform.

**Data Availability**

Our data are archived with the British Oceanographic Date Centre (BODC, https://www.bodc.ac.uk/), the bubble data are available at DOI: 10.5285/c972e316-2b93-1b4e-e053-6c86abc02285 and the wave data can be found at doi: 10.5285/c9ae04d6-32d2-73f1-e053-6c86abc0c833. Other HiWinGS cruise data, including the near-surface meteorology used here are available from: ftp1.esrl.noaa.gov/psd3/cruises/HIWINGS_2013/Collective_Archive.


**Author Contribution**

HC and IMB designed the bubble and wave state section of the HiWINGS project and operated all sensors at sea. BB was the Chief Scientist of HiWINGS, and oversaw all operations. RP built the buoy and was responsible for buoy operations at sea. SG designed and built the bubble camera and analysis software. AM helped with deployment at sea and carried out the sonar

analysis. HC was responsible for the bubble sensor deployment and carried out analysis of all bubble data except the sonar. IMB performed the 2D wave analysis. HC and IMB prepared the manuscript.

**Competing Interests**

The authors declare that they have no conflict of interest.




**Acknowledgements**

This work was funded by the Natural Environment Research Council under grants NE/J022373/1 (HC) and NE/J020893/1 (IMB), NE/J020540/1 (NOC) and HC's fellowship NE/H016856/1. Funding for ship time was provided under US National Science Foundation grant AGS 1036062. We are grateful to the captain and crew of the R/V *Knorr* for their invaluable

assistance at sea. We gratefully acknowledge the contribution of Nick Hall-Patch & Svein Vagle for their work on the construction and operation of the acoustical resonators used.

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





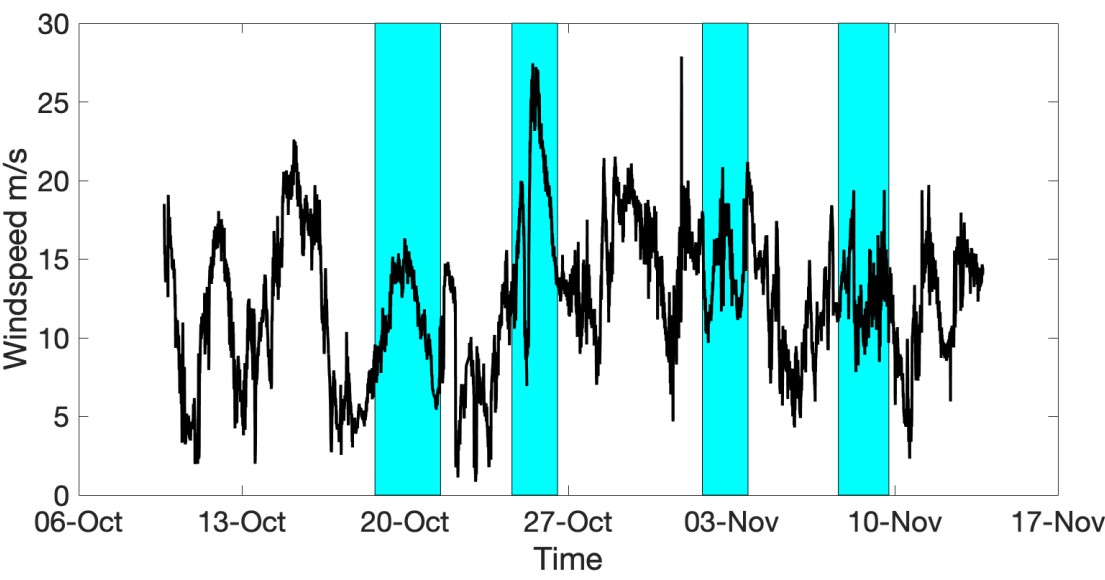

.030

**Figure 1: Wind speed over the entire HiWINGS expedition. Shaded areas show the times of the four buoy deployments discussed in this paper.**

.035

.040

.045



| | Station 3 | Station 4 | Station 6 | Station 7 |
|---|---|---|---|---|
| Period of measurement | 18-Oct-2013 17:00- 21-Oct-2013 12:30 | 24-Oct-2013 14:00:00 - 26-Oct-2013 13:00:00 | 01-Nov-2013 18:00:00 - 03-Nov-2013 17:00:00 | 07-Nov-2013 14:00:00 - 09-Nov-2013 18:00:00 |
| Total measurement time | Camera: 3.75h<br><br>Resonator: 6h<br><br>Sonar: 35h (cont) | Camera: 8.25 h<br><br>Resonator: -<br><br>Sonar: 26 h (cont) | Camera: 9h<br><br>Resonator: 35 h (cont)<br><br>Sonar: 36 h (cont) | Camera: 8.25 h<br><br>Resonator: 11 h (cont)<br><br>Sonar: 38 h (cont) |
| Sea surface temp. | 8.7 C | 10.2 C | 8.7 C | 20 C |
| Wind speed range | 5.9 - 15.3 m/s | 7.7 - 26.6 m/s | 10.5 - 18.7 m/s | 9.5 - 18.0 m/s |
| Sig. Wave height range | 1.7 - 5.3 m | 3.0 - 11.0 m | 3.0 - 5.0 m | 2.2 - 5.0 m |
| Start position | 54.1 N 46 W | 53.5 N   45.4 W | 52.0 N  50.0 W | 41.45 N   64.0 W |
| Mixed layer Depth | 30 - 50 m | 50 - 70 m | 60 - 75 m | 30 - 70 m |
| Mean ADV depth | 3.47 +/- 0.19 m | 4.13 +/- 0.39 m | 3.96 +/- 0.21 m | 3.90 +/- 0.19 m |
| Chlorophyll | 1.2 - 1.7 $\mu$g/l | 0.6 - 1.2 $\mu$g/l | 0.5 - 0.9 $\mu$g/l | 0.3 - 0.8 $\mu$g/l |

**Table 1: Wind speed over the entire HiWINGS expedition. Shaded areas show the times of the four buoy deployments discussed in this paper.**




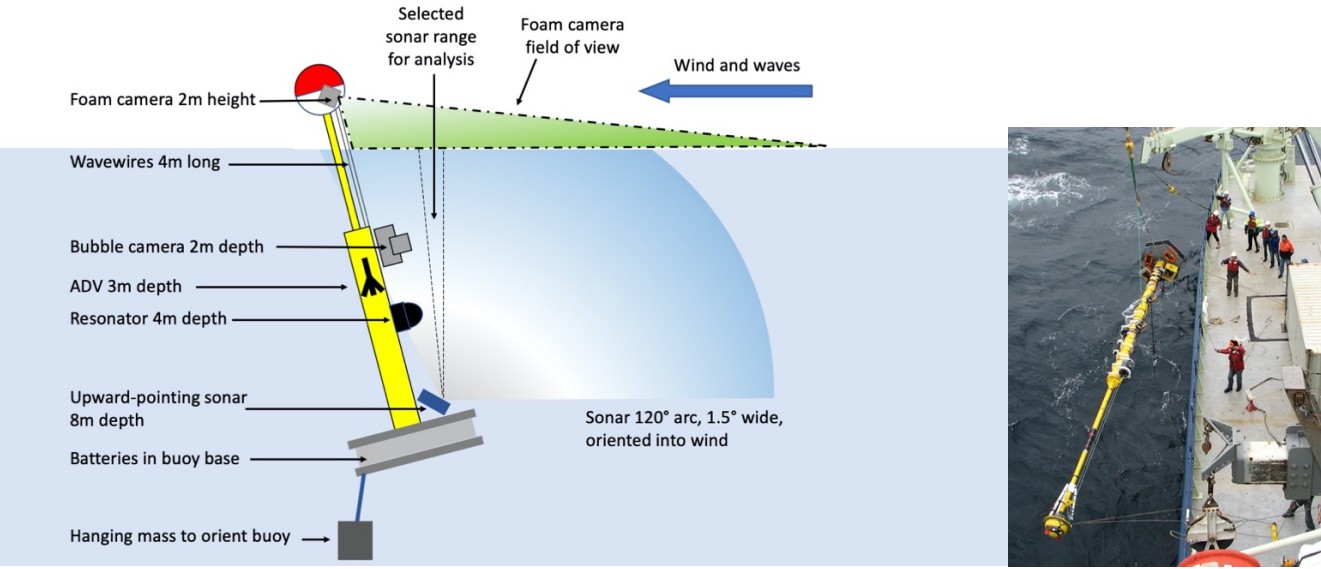

.050

**Figure 2: (a) Schematic diagram of the 11-m long spar buoy. The top 2 m protruded above the water level, and the hanging mass caused a slight backwards lean which oriented the buoy into the wind. All instruments were positioned on the upwind side except the ADV which was 90 degrees further round the hull. The sonar data presented here is a subset of the full sonar arc, a 10° vertical section shown between the dotted lines and treated as a vertical profile through the water. (b) Buoy being deployed.**

.055    .





**Figure 3:** Comparison of simultaneous measurements from (a) the upward-looking sonar, (b) the bubble camera at 2 m depth, and (c) the resonator at 4 m depth. These plots cover one 45-minute measurement period of the bubble camera, on November 2$^{nd}$ from 1805-1850h. The wind speed was 18 ms$^{-1}$. (a) shows backscatter cross-section per unit volume in dB from a vertical 10° section of the arc; the dotted lines show the vertical position of the camera and resonator. To make the comparison between instruments clear, the sonar data is shown relative to the equilibrium water surface position on the buoy, not adjusted to instantaneous depth as waves pass. (b) and (c) show void fraction for camera and resonator respectively, on a linear scale on left, and a log scale on right. Note that the y-axis scales for camera and resonator differ by two orders of magnitude. There is good agreement in the features observed between all three instruments.

.





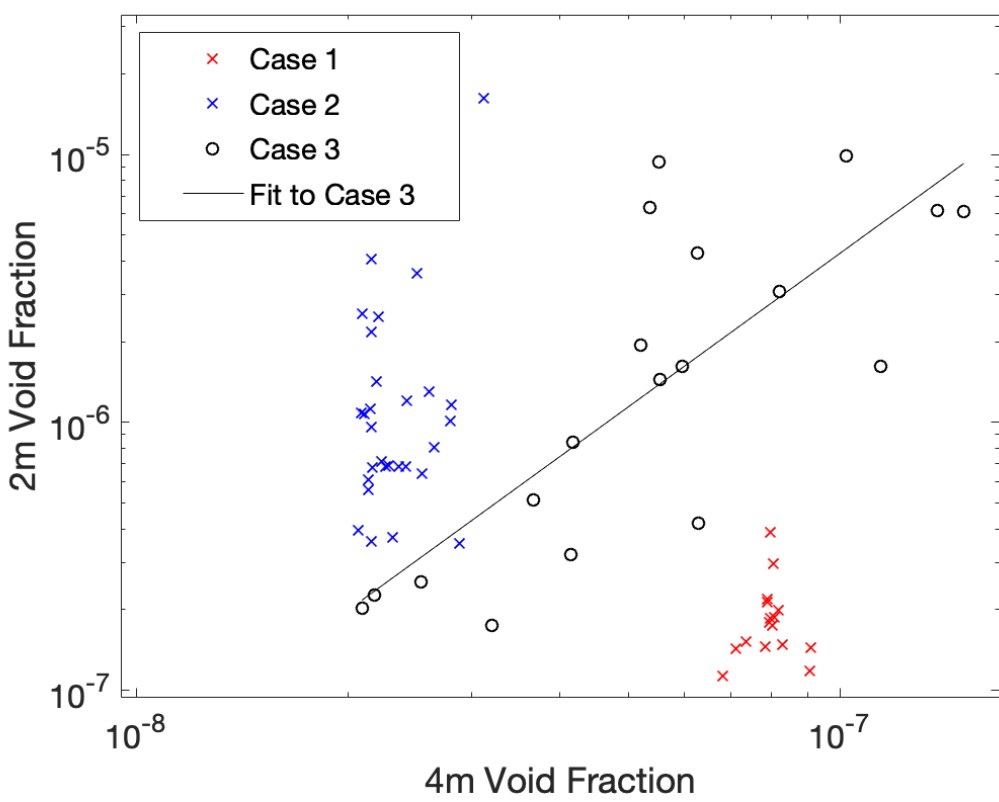

.070

**Figure 4:** Scatter plot comparing one minute average void fractions at 2 m and 4 m during station 6 (1–4 November). It shows all periods when the data for both camera and resonator were above the noise level. The data is split into three cases: (1) very few 4 m bubbles but a high void fraction at 2 m; (2) very few at 2 m but a high void fraction at 4 m; and (3) the cases where they approximately .075 match.

.





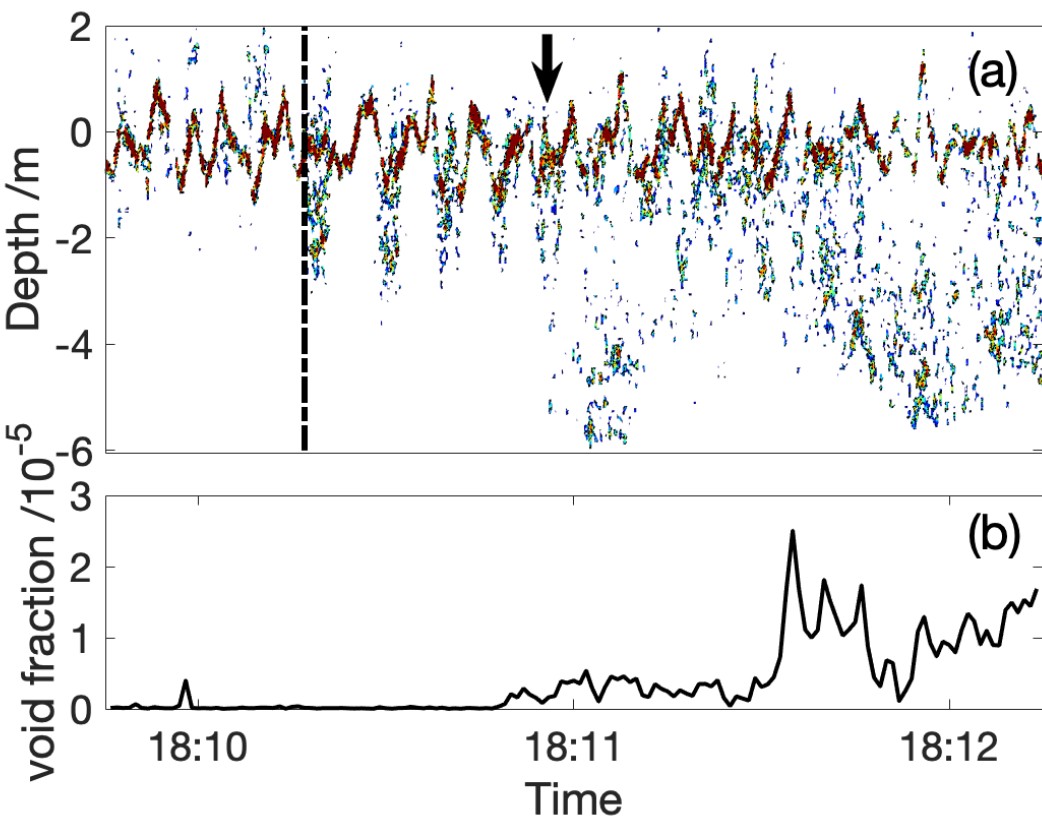

Figure 5: Figure (a) shows a contour plot of sonar backscatter during a 150s period on November 2$^{nd}$ from 18:10h – 18:12h.  The red colour indicates the strong reflections from the instantaneous sea surface, showing the passing waves.  The vertical dotted line shows the time that the whitecap was observed in the foam camera images which coincides with the appearance of a shallow bubble plume. The bold arrow is aligned with the rapid deepening of the bubble plume approximately 40 seconds later. (b) shows the void fraction observed by the submerged bubble camera (at 2 m) during the same period.

.080

.085    .







**Figure 6: Running average flow speeds (averaged over 16 seconds, which is two periods of natural vertical oscillation for the buoy)**

.090 **in (a) x, (b) y and (c) z directions. The box indicates the time segment which corresponds with the sonar data in figure 5. 'u' indicates the direction into the wind, v is the sideways axis to the port side and w is the vertical direction, all in the buoy frame of reference. The dotted lines show the average of flow speed over this period (which are non-zero because of Stokes drift in the u direction and the buoy lean in the w direction). There is a notable sideways flow feature as the plume forms, and the vertical velocity at that time is generally downward, possibly indicating surface convergence in a Langmuir circulation flow pattern.**

.095 .




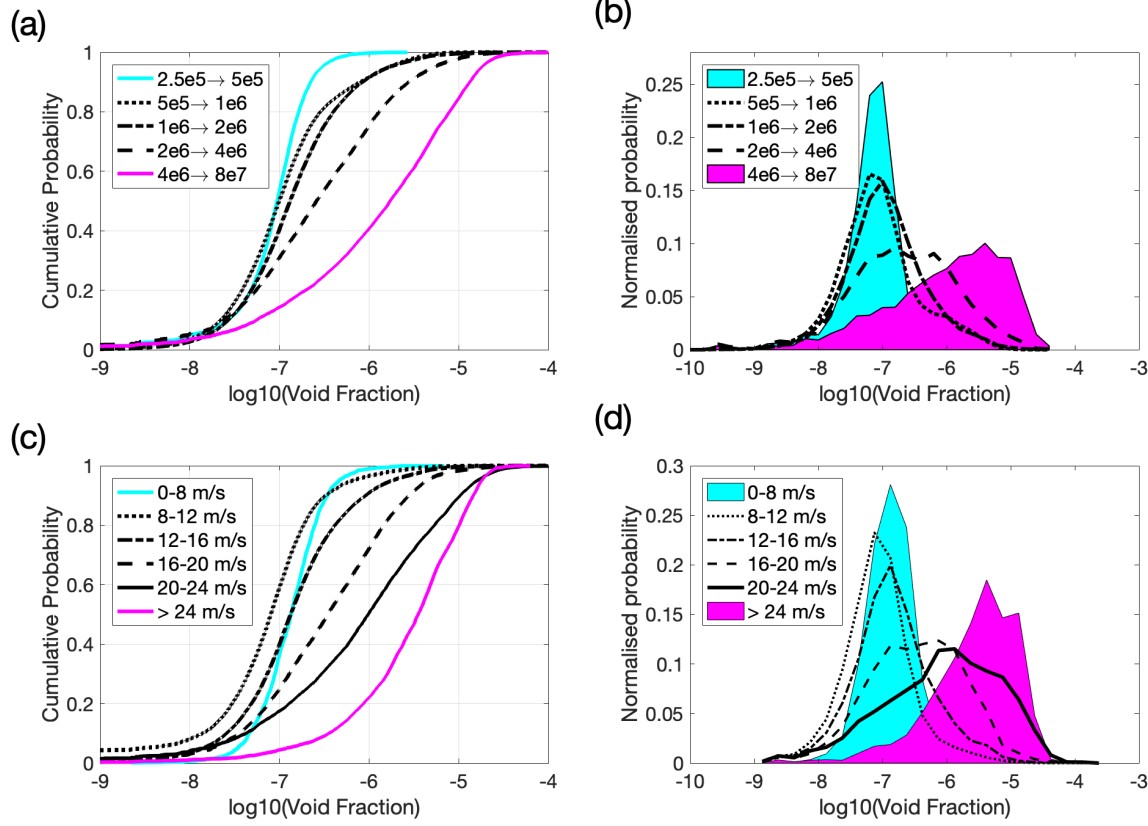

**Figure 7: 1-second averaged void fraction probability distributions at 2 m depth, split by wind speed and wind-wave Reynolds number. All the data collected by the bubble camera during all deployments is included in these plots. (a) and (c) show cumulative distributions and (b) and (d) show the same data as normalised distributions. For (b) and (d), the shaded regions show the lowest and highest values of the range. Wind speed and Reynolds number are calculated for ten minute periods, using the 10-minute mean for $U_{10}$ and the interpolated windsea-only significant wave height. We note that although there are data for windspeeds of 0-8 m s[-1], these are all calm periods just after much higher winds.**





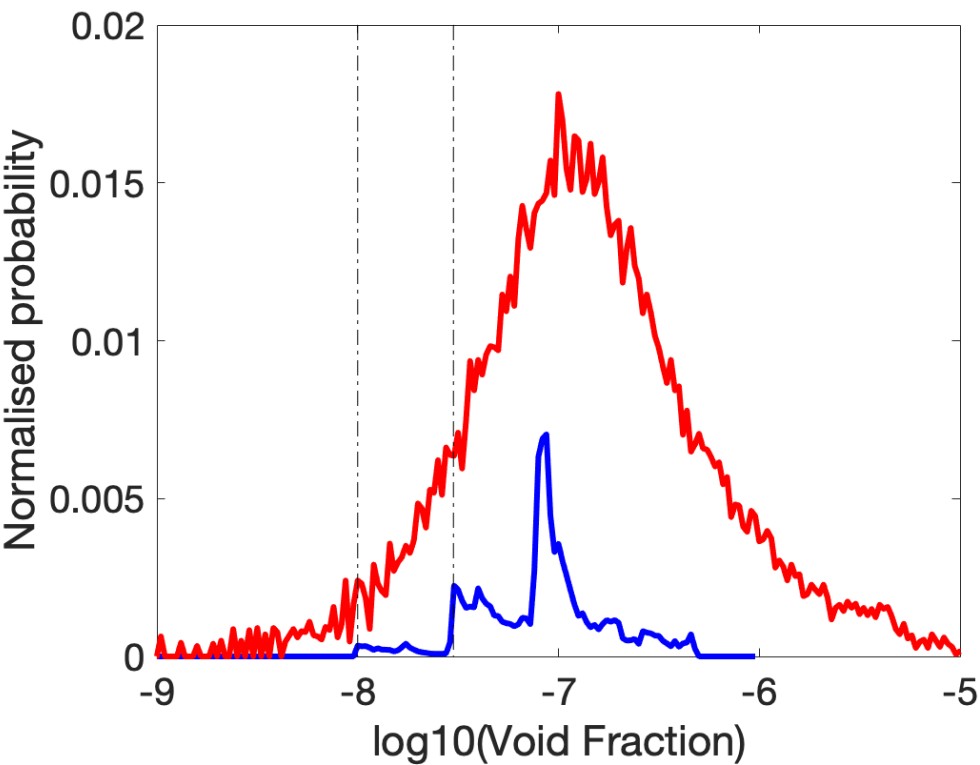

.110 **Figure 8: Normalised void fraction probabilities at 4 m depth (blue) compared with the equivalent normalised probabilities at 2 m depth (red). The dotted lines are the noise levels for station 3 ($1{\times}10^{-8}$) and station 6 ($3{\times}10^{-8}$). The final deployment is not included because it yielded so little data. This plot only shows data collected while the wind-wave Reynolds number was between $10^6$ and $2{\times}10^6$, and narrow bins are used to enable examination of the peak positions. The resonator data has been normalised to include the 90% of the time that the measured void fraction did not rise above the noise (and the data below the noise level is not represented**
.115 **here).**

.




**Figure 9: Median void fraction in each 2 m s⁻¹ wind speed bin for all deployments. In (a), these bins are split into periods of rising and falling winds. In (b) the same data is split by inverse wave age, representing developing (>0.04), and mature seas (<0.036). Shading represents the standard deviation at each wind speed. There were bubble camera data for mature seas when the wind speed was greater than 19 m s⁻¹. (c) shows the resonator data separated by wind speed. There is insufficient resonator data for a comparison with inverse wave age.**





**Figure 10: A typical offset between void fraction peaks at depths of 2m (a) and 4m (b). The plume structure is very similar, although the void fractions in the two cases differ by two orders of magnitude and there is a clear offset of 25 seconds. (c) shows the offset distance in metres for compared to the ratio between the peak void fraction and 2m and 4m. Marker colour shows log$_{10}$(peak void fraction) of the camera measurement for each pair. All measured pairs are shown with the exception of two which had offsets of -4.8m and 27m. (d) Windspeed at the time the offset was measured. Marker colour is the same as (c).**

.130

.

.135




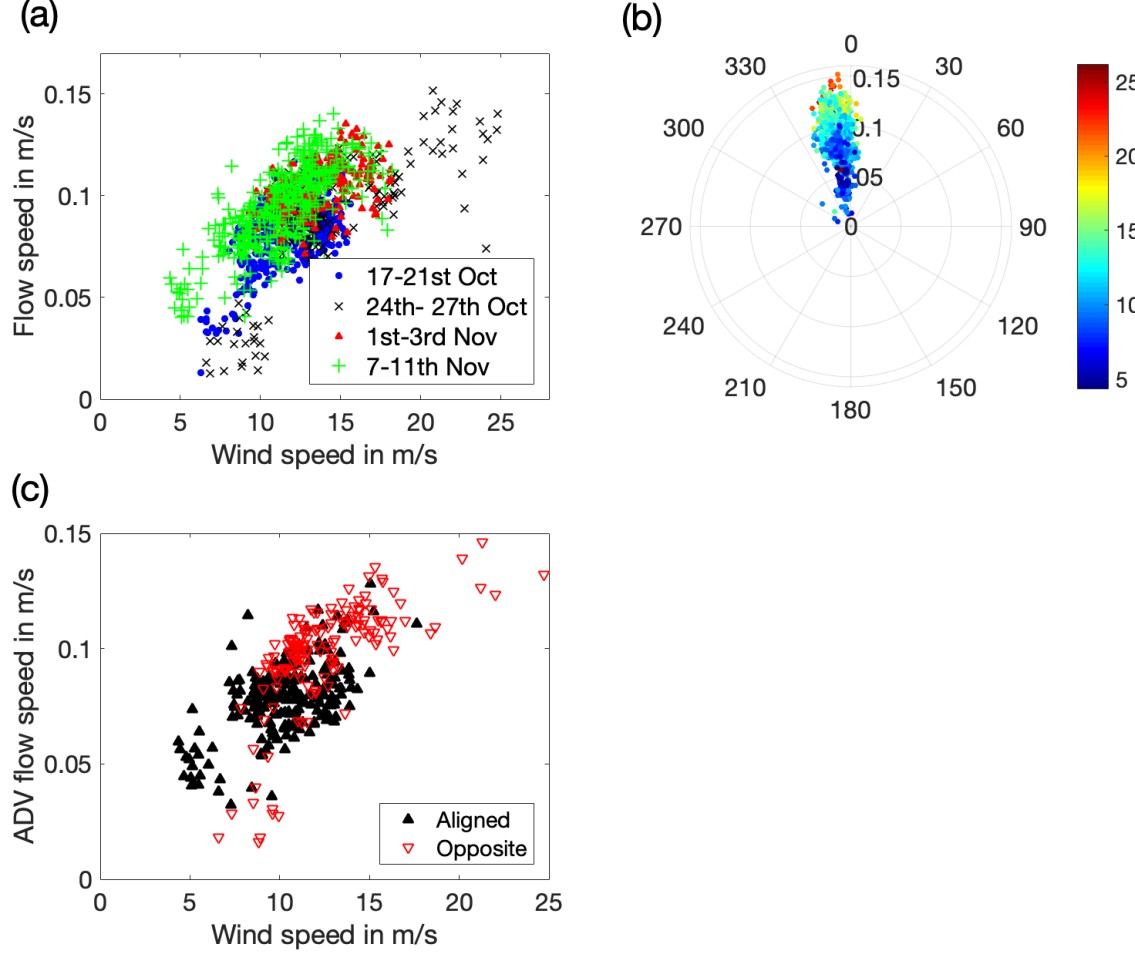

**Figure A1: Ten minute averages of ADV data showing water flow relative to the buoy at 3.8m depth. (a) Horizontal flow speed. (b) The direction of the measured water velocity relative to the buoy (0° is the upwind direction). The radial parameter is the total measured flow speed in m.s⁻¹, and markers are colour-coded by wind speed in m.s⁻¹. (c) Selected data from periods when the swell direction was within +/- 45° of the wind ("aligned") and within +/- 45° against the wind ("opposite").**

.140

.




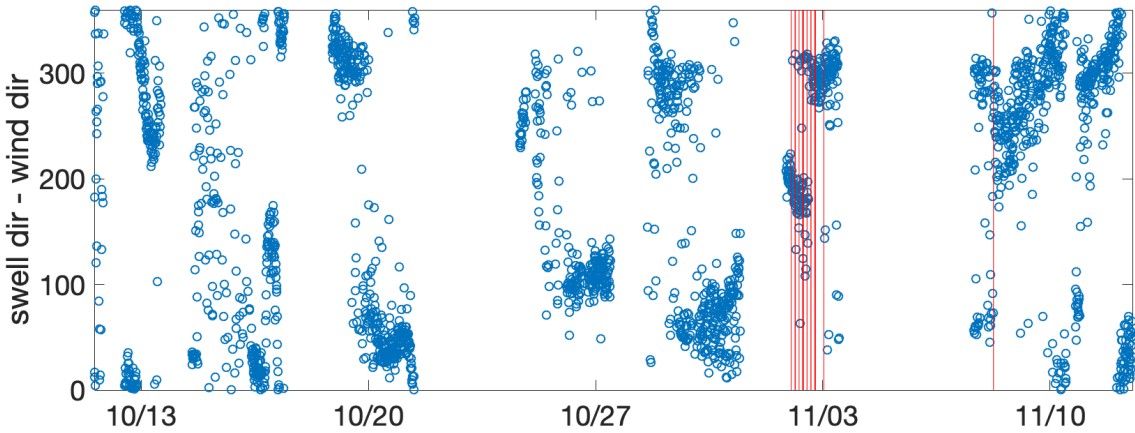

.145
**Figure A2: The angular difference between ten minute averages of the wind direction and the direction of the dominant swell. The red lines show the times when measurements of the time difference between plume appearance at 2 m and 4 m were taken. No offset measurements were possible for the deployment between the 24th-27th of October because no resonator data is available.**

.

.150

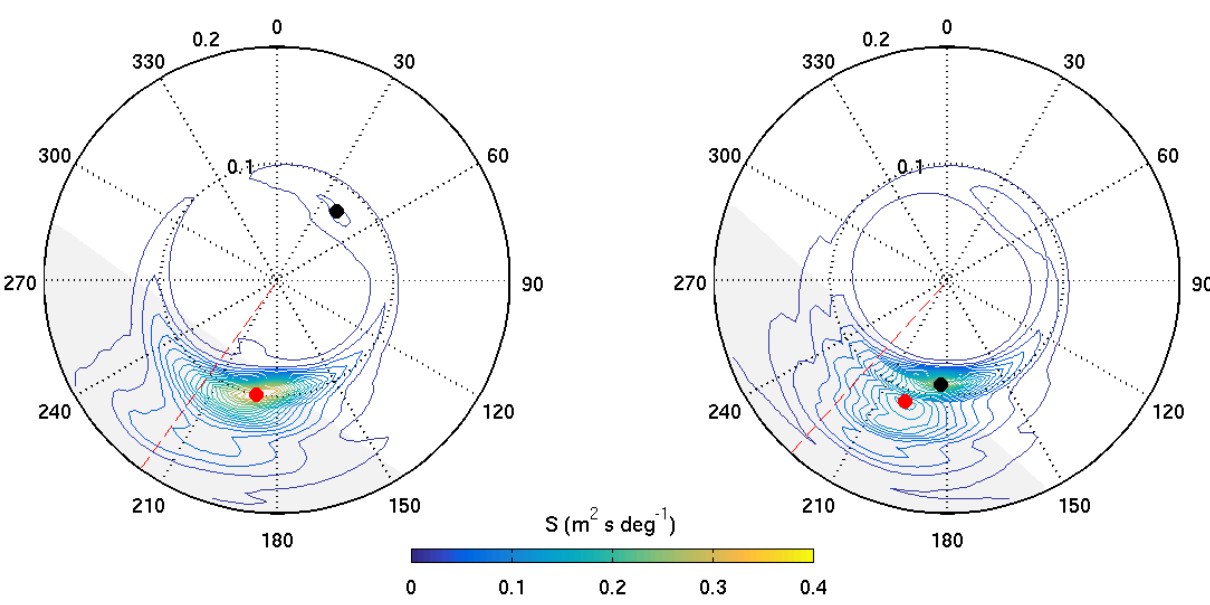

.155
**Figure A3: Observed 2D directional wave spectra measured by the WaveRider at (a) 12:45 and (b) 13:15 on November 2nd. The radial parameter is frequency (Hz), 0 degrees is north, and the contour lines show spectral intensity. Red circles show the main wind sea peak identified by the algorithm and black circles show the identified dominant swell. At 12:45, $Hs_{windsea}$ = 4.26 m and $Hs_{swell}$ = 1.25 m.**



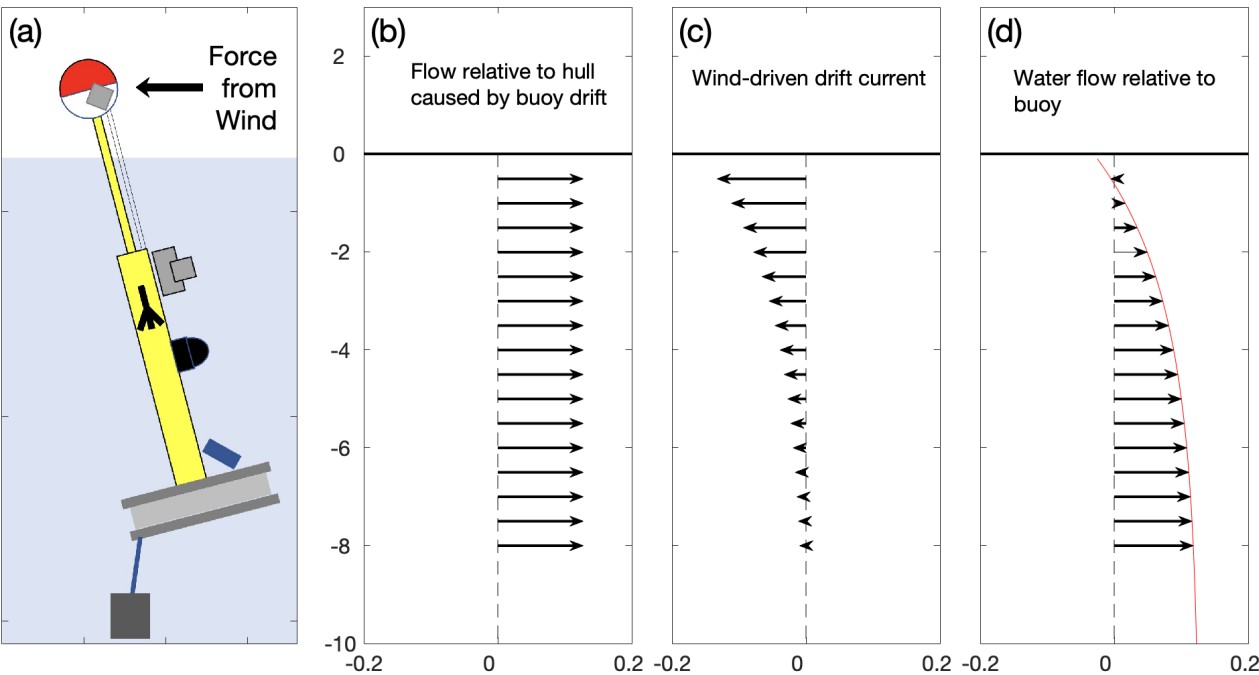

.160    **Figure A4: The combination of influences contributing to the water flow profile in the frame of reference of the buoy. (a) shows the buoy oriented into the wind, with the dome acting as an obstacle to the wind and transferring horizontal force to the submerged hull. (b) shows the effect of flow speed relative to the buoy caused by the buoy moving through the water in response to a 15 m s$^{-1}$ wind. Arrow length represents water flow speed and the magnitude is shown on the x-axis in m s$^{-1}$.  (c) shows the current due to Stokes drift, calculated as described in the main text. (d) shows the combination of (b) and (c) relative to the buoy hull. The red line**
.165    **shows the complete profile.**

.170