# Peer review of "Ocean bubbles under high wind conditions. Part 1: Bubble distribution and development"

_Ocean Science, 2021_

## Author Comment (AC2)

We are grateful to the reviewer for their comments and have prepared a revised manuscript which addresses their concerns. Our response to each point and a description of the changes made are given below (reviewer comment in blue, author response in black).

**Scientific issues:**

**Lin 334, Figure 4: Splitting the data into 3 cases seems very arbitrary. Give more detail on how the data were assigned to a particular case. For example, why are the Case 2 data with 2m void fraction < 7 x 10 -7 not part of Case 3. If there is no objective way to stratify the data, then present them as one case and a fit through all data – or best to remove this figure.**

The previous separation was based on data points either both meeting or both falling below numerical criteria. We agree that this appears arbitrary and in the revised manuscript we have removed the case distinction. However have left the figure in because we think it makes an important point. We have noticed a general assumption in the scientific community that bubble plumes vary reasonably smoothly with depth (which is the impression given by sonar backscatter data in other papers) and we think that it is important to show the extent of the heterogeneity at small local scales.

**Ln352: What is the area of the camera FOV? Please quantify "very few ". How does the observed number of breaking waves compare to the breaking rate reported in the literature? (E.g. there are several studies that quantify the active breaking rate from breaking crest length distributions obtained from video imagery). If the observed breaking rate is much lower than expected, it might indicate that the freely drifting buoy somehow avoids breaking waves., similar to wave buoys often missing the peak of a steep crest, or by the buoy being located preferentially in the convergence region of Langmuir circulations. If so, the bubble statistics would be biased.**

The camera had a fisheye lens but was only 2m above the ocean surface, and so the effective imaging footprint only extended 3-4 metres upwind of the buoy. The camera position was chosen to see the detail of waves breaking just in front of the buoy to correlate with the subsurface measurements, and was not designed to provide data on breaking statistics. The field of view was too small, given the low probability of waves breaking directly within a specific 4m x 4m patch of ocean. "Very few" here is less than ten over the whole campaign, and so we cannot compare breaking statistics. We have added text to the manuscript to clarify this point, and a reference to a paper that has some description of the setup and and example pictures. The buoy behaviour was observed carefully while it was within sight of the ship, and we saw no evidence that the buoy position was biased.

**Ln405: Can you exclude the possibility of measurement saturation at high void fractions? Turbulence models assume TKE injection in breaking waves to a depth comparable to the**

**significant wave height. It is surprising that bubbles would not get injected to 2m depth when Hs is 3 to 5m. Does this imply that void fractions in breaking waves are <10-4? Or does it mean that the breaking layer is << Hs?**

The bubble camera and resonator have both been extensively tested and were designed for void fractions of $10^{-4}$ and above. The companion paper discusses the bubble size distributions at these high void fractions and there is no evidence of bias. We are confident that there is no saturation of the measurement, and we have added text to clarify this point in the manuscript. We have GoPro footage of breaking waves (not shown in this paper, but from this expedition at ~15 m/s) does not show visible bubbles from a breaking wave mixing down below the top half metre or so – they stay in a very shallow layer. We also note that the bubble size distributions that reach 2 m depth (discussed in the companion paper) show that bubble populations that reach 2m depth are already highly processed within the top metre and are much smaller than turbulence arguments alone would suggest.

Although turbulence models assume a depth of TKE injection comparable to the significant wave height, we are not aware of any experimental data collected at sea to support the idea that bubbles are injected to this depth. The breakers on the open ocean are spilling rather than plunging and bubbles are formed on the front wave face with forward rather than downward momentum. Our conclusion is that the breaking layer is << Hs, and we discuss this in more detail in the companion paper.

**Section 3.3.2: Your analysis shows that void fraction does only poorly stratify with wind history. In almost all cases the median void fraction for rising/falling winds is within 1 standard deviation of the opposite situation. Similarly, wave age does not affect void fraction. Both results are not surprising. Void fraction is ultimately determined by wave breaking, which in turn is not directly linked to wave age or wind history. But is well correlated with the saturation of the wave field (see work by M. Banner and colleagues).**

The literature has relatively few direct bubble measurements in high winds close to the sea surface to constrain detailed mechanisms for bubble distribution and gas flux. Our aim here was to examine the possible influence of wind history on void fraction patterns at 2m depth, since the mechanism of deep plume formation by Langmuir circulation (which we discuss further in the companion paper) does not necessarily correlate directly with the bubble population details in the top metre of the ocean. As we set out in the companion paper, we think that there are several linked mechanisms influencing bubble distribution, but some of these mechanisms may have a stronger influence on gas fluxes than others. Recent papers (for example Liang 2017) suggest that rising and falling winds have different effects on bubble distribution and gas fluxes, and think that the direct observations are useful even if they do not show strong separation between cases.

**Ln 517: Can the (relative) age of a bubble plume be determined from the bubble size distribution? If this is done in the companion paper it should be mentioned here. If not, the analysis should be included here.**

We have maintained the separation of material in these papers (void fractions in this first paper, bubble size distributions in the second) because there is too much material to fit into one paper and because both data types have their own nuances which need significant discussion in context. One major finding in the second paper is that the bubble size distribution at 2m for void fractions above $10^{-6}$ is highly conserved, varying in void fraction due to dilution but with the same relative numbers of bubbles of different sizes. There is no evidence at 2m depth of either buoyancy sorting the bubbles over time (with the exception of a small number of very large bubbles) or size-dependent

dissolution. Our conclusion is that these fixed populations are formed within the top metre and are only redistributed through advection or destroyed through size-independent collapse after that. So the relative age of bubble plumes can not be determined by the bubble size distribution at 2m, but we do not think it appropriate to include that analysis here because this is a nuanced point and the details of the bubble size distributions are the subject of considerable discussion in the second paper.

**Ln594-619: The explanation of higher void fractions during decreasing wind speeds being the result of higher gas saturation is plausible. Another explanation is similarly plausible: during falling wind speeds the wave field is not equilibrated, resulting in a higher breaking rate (bubble production) associated with wave saturation at a prior time.**

We have added this hypothesis to the manuscript (lines 464-467).

**Technical issues:**

**Ln 24: define $R_{Hw}$**

We have added the definition in the revised version, with text but not an equation, so it now states "… or a wind-wave Reynolds number of $R_{Hw} = 2 \times 10^{6}$,…".

**Ln 159: change to: "Studies by Salter et al (2014) and Slauenwhite & Johnson (1999) observed…."**

This change has been made in the revised manuscript.

**Ln294: The standard deviation of the measurement depth is not dimensionless (add units to both instances).**

Extra text has been added to clarify the units.

**Figure 3: Change y-axis label to "Void fraction" on panel b and c**

This has been changed in the revised manuscript.

**Figure 7: Specify in caption "….split by wind speed (bottom row) and wind-wave Reynolds number (top row)."**

This change has been made in the revised manuscript.

**Figure9, caption Ln123: "There were no bubble camera data…"**

This change has been made in the revised manuscript.

---

## Author Comment (AC3)

**Response to Referee #2**
**"Ocean bubbles under high wind conditions. Part 1: Bubble distribution and development" by Helen Czerski et al., Ocean Sci. Discuss.,**
**https://doi.org/10.5194/os-2021-103-RC2, 2021**
**27th January 2022**

We are grateful to the reviewer for their comments and have prepared a revised manuscript which addresses their concerns. Our response to each point and a description of the changes made are given below (reviewer comment in blue, author response in black).

**Panel (a) of Figure 3 indicates that bubble ~ 12 microns could go down to 6 m at a wind speed of 18 m/s. I recall backscatter data at ocean station Papa (by Svein Vagle) showing that bubbles of around 20 to 30 microns go down to more than 10 meters at a wind speed of ~10 m/s. I believe a figure showing the data is in the textbook "Chemical Oceanography and the Marine Carbon Cycle" by Steve Emerson (I apologize that I do not have the book with me right now and do not have the figure number). Do bubbles of 12 microns dissolve quickly in the water column and do not exist below that depth? Or is there any limitation in the observation that bubbles deeper than 6 m cannot be detected?**

We think that the figure you refer to is Figure 10.9 in Chapter 10 of the Emerson book (although there is a spelling mistake in SV's name in the text). This figure shows data from a 120 kHz sonar, which would be resonant with bubbles of 27 microns in radius (detectable by both camera and resonator in our experiment). The wind speed given is 12 m/s. It shows backscatter intensity contours between -35 and -70 dB, and although the -70dB contours do reach to 10 metres, the ones at -45 dB (comparable with our data, see the scale on Figure 3) are much closer to ours, showing a contour that rarely reaches beyond 5m. We also note that the Vagle data is averaged over a period of 2.25 hours, whereas ours are 1 and 10 second averages. The way the Vagle data is processed would allow extremely small numbers of resonant bubbles at depths of 10 m to appear to create deep plumes. We think that the data in the Vagle plot is reasonably consistent with ours if the backscatter intensity (making their detection more sensitive) and time-averaging are taken into account, although they do appear to see slightly deeper bubbles at 12 m/s. There is a more in-depth discussion of similar data (but with a 200 kHz sonar) in the Vagle 2010 paper that we cite, which also shows contours that are in general agreement with ours at the same backscatter intensity level. We have added a note on this comparison to the discussion section (line 330). We note that we were sampling at a single location and it's not clear how similar bubble behaviour will be in very different areas of the ocean (with different temperatures, mixed layer depths and gas saturation levels), and that therefore a reasonable similarity between these two sites is notable.

**Also for Figure3: I would suggest revising the ylabel for panels b and c as "void fraction". The current ylabels seem to be more appropriate as titles.**

This has been changed in the revised manuscript.

**Regarding the results in Figure 4: Figure 10 shows that the void fraction at 4 m could lag behind that at 2 m. Is that already considered in the correlation presented in Figure 4?**

The data in figure 4 are one minute averages (stated in the caption) and the lags shown in Figure 10 are between 0 and 66 seconds (stated in the main text). The majority of the lags were 30 seconds or

less and would have fallen within the same 1 minute averaging period. We have looked again at the data and conclude that the timing mismatches on the scatter plot are not the main influence on the lack of correlation on the scatter plot, partly because of the time averaging and partly because the data in Figure 4 includes all periods when the void fraction at 4 m was above the noise whereas the data in Figure 10 only shows the peaks (20 points, each at the peak of that plume).  The point that we wish to make here is that the plume structure is locally heterogeneous, rather than to discuss specific causes.  We consider that at least some of the mismatch between void fractions at 2m and 4m is likely to be caused by shear currents over longer periods of time, although other advection processes are also likely to contribute.  We have added text to clarify this in the manuscript.

**Currents in Figure 6 are useful to explain void fractions in Figure 5. I would suggest the author also try to connect Figure 6 to explain Figure 3. For example, the downward current at around 18:30 is strong than at around 18:10, but the increase in the void fraction is smaller. Is it because there is no breaking wave observed at 18:30?**

Our full description of the mechanisms we propose is given in the companion paper, and there we suggest that the heterogeneous surface layer (the top 1m) is created by waves breaking randomly across the surface, but that it is only where this layer is pulled downward by Langmuir circulations that a deep plume can form. However, there can only be a plume if there were bubbles in the upper layer before it was advected downwards.  The composite figure showing sidescan sonar data from Zedel & Farmer (figure 9 in their 1991 paper) shows long heterogeneous bubble plumes with large scale patterns evident but considerable local variation, which is consistent with what we see.  We have added a comment about the comparison between Figure 3 and Figure 6, and the implications for heterogeneity on lines 375-380.

**Although this is mentioned in the text, I would suggest adding that the measurement is at 3.8 m in the caption of Figure 6.**

This has been added in the revised manuscript.

**Figure 9: I would suggest the authors clarify if the rising/falling wind means a sustained period of rising/falling wind, like in Liang et al. 2017 JGR-Oceans referenced in the manuscript. Or do the authors include substantial periods when the wind is fluctuating?**

The manuscript (lines 448 – 450) describes the distinction: "*If the hourly averaged wind speed was lower or higher than the mean of the previous two hours by 0.5 m/s, the data from that hour was labelled "falling" or "rising" respectively.  Approximately one quarter of the data fell into each of those two categories.  Otherwise, data points were considered ambiguous and are excluded.*".  This is the only method used to separate the two categories.  Periods of fluctuating wind are likely to be considered ambiguous and will therefore be excluded.  We do not use the broad definition adopted by Liang et al, which compared a single 24h period of steeply rising winds to a separate 36h period of steeply falling winds. We have added brief text to the caption of figure 9 to clarify that the method used to separate rising and falling winds is in the main text.

**The data in Figure 9(a) show that void fraction is higher at falling wind than at rising wind until about 20 m/s. This could be interesting results and I have different thoughts from the paragraph starting from line 456. Breaking waves are bigger at the rising wind, but Langmuir turbulence is stronger during the falling wind. I wonder if there is a possibility that void fraction at this depth is primarily due to Langmuir circulations at wind speed < 20 m/s and gets more contribution from breaking waves when wind speed > 20 m/s. The same argument could be used to explain that void fraction is mostly larger during**

**falling wind than during rising wind at 4 m. However, I could not make sense of why the void fraction is the largest when wind speed is at 10 m/s and 12.5 m/s. Is there any sampling error there?**

We have added this potential explanation to the manuscript (lines 464 – 467).  We are confident that there is no sampling error, but we feel that our original suggestion of hysteresis at the extremes of the wind speed range can explain the situation at the low wind speeds.  We also note that a relatively small fraction of the data set was collected at the very lowest wind speeds, and so it may have less statistical significance.

**Regarding the presentation of Figure 9. I would suggest stating in the caption that panel a is from data at 2 m and panel c is from data at 4 m.**

This has been added in the revised manuscript.

**Figure 9(b) is referenced after Figure 9(c). Perhaps the order of the panels could be changed.**

We have chosen not to do this because we think it's more intuitive to compare figures (a) and (c) if they are directly above one another so that they effectively have a common x-axis.  We are happy to make this change if the reviewer insists on it.

**Both "parametrization" and "parameterization" are used. I would suggest the authors pick one of them.**

The Oxford English Dictionary and papers from our colleagues in this area suggest "parameterization", so we have checked that this usage is now consistent throughout the revised manuscript.